# A Fused Gromov-Wasserstein Approach to Subgraph Contrastive Learning

**Amadou S. Sangare**  *amadou.sangare@telecom-paris.fr*
*LTCI, Télécom Paris*
*Institut Polytechnique de Paris, France*

**Nicolas Dunou**  *nicolas.dunou@mines-saint-etienne.org*
*École des Mines de Saint-Étienne*
*Université Paris Dauphine-PSL, France*

**Jhony H. Giraldo**  *jhony.giraldo@telecom-paris.fr*
*LTCI, Télécom Paris*
*Institut Polytechnique de Paris, France*

**Fragkiskos D. Malliaros**  *fragkiskos.malliaros@centralesupelec.fr*
*Université Paris-Saclay*
*CentraleSupélec, Inria, France*

**Reviewed on OpenReview:** *https://openreview.net/forum?id=J7cY9Jr9WM*

## Abstract

Self-supervised learning has become a key method for training deep learning models when labeled data is scarce or unavailable. While graph machine learning holds great promise across various domains, the design of effective pretext tasks for self-supervised graph representation learning remains challenging. Contrastive learning, a popular approach in graph self-supervised learning, leverages positive and negative pairs to compute a contrastive loss function. However, current graph contrastive learning methods often struggle to fully use structural patterns and node similarities. To address these issues, we present a new method called **F**used **Gr**omov **Wa**sserstein **S**ubgraph Contrast**i**ve **L**earning (FOSSIL). Our model integrates node-level and subgraph-level contrastive learning, seamlessly combining a standard node-level contrastive loss with the Fused Gromov-Wasserstein distance. This combination helps our method capture both node features and graph structure together. Importantly, our approach works well with both homophilic and heterophilic graphs and can dynamically create views for generating positive and negative pairs. Through extensive experiments on benchmark graph datasets, we show that FOSSIL outperforms or achieves competitive performance compared to current state-of-the-art methods.

## 1 Introduction

Self-Supervised Learning (SSL) is a methodology employed in machine learning to extract meaningful and transferable representations from unlabeled data. SSL has been successfully used in many areas, like computer vision (Tomasev et al., 2022), speech processing (Baevski et al., 2020), and natural language processing (Devlin et al., 2019; Lewis et al., 2019). SSL is particularly important when labeled data is impractical or costly to obtain. Graph Representation Learning (GRL) stands out as a domain where SSL excels due to the complexity involved in annotating graphs, which often demands specialized knowledge and cannot be easily crowd-sourced. For instance, in biochemistry (Zang et al., 2023), annotating molecular graphs requires specialized knowledge, significantly increasing the labeling effort compared to other data modalities.

In SSL, Contrastive Learning (CL) is a prominent approach (Chen et al., 2020b). CL methods typically involve sampling an anchor from the dataset, generating an augmentation (positive pair), and contrasting it with negative pairs using a CL loss function. Unlike certain modalities, such as images, the generation of positive pairs in GRL presents unique challenges. Various methods have been proposed, including those involving node-level or subgraph-level contrastive loss functions (Zhu et al., 2020b; Yuan et al., 2023; Han et al., 2022; Jiao et al., 2020). Recent studies have shown that subgraph-level approaches outperform node-level ones, as they can capture the structural similarities between the anchor and positive graph pair (Han et al., 2022).

Prior subgraph-level CL methods often use readout functions to extract subgraph representations (Jiao et al., 2020; Veličković et al., 2019). Alternatively, some approaches rely on Optimal Transport (OT) distances for computing the CL loss (Han et al., 2022). OT-based techniques exhibit superior performance compared to readout-based methods in subgraph-level CL. Notably, the Generative Subgraph Contrast (GSC) method (Han et al., 2022) leverages a combination of the well-established Wasserstein Distance (WD) (Villani et al., 2009) and Gromov-Wasserstein Distance (GWD) (Mémoli, 2011) from OT. However, existing OT-based methodologies face two challenges: separately computing the WD and GWD does not capture both structural and feature information simultaneously when comparing subgraphs, and the Graph Neural Network (GNN) encoder struggles with heterophilic datasets.

Motivated by the limitations of previous subgraph CL methodologies, we introduce a new GCL approach employing the Fused Gromov-Wasserstein Distance (FGWD) (Vayer et al., 2018; Brogat-Motte et al., 2022) from OT. The FGWD captures both feature and structural similarities jointly. Our model, termed **F**used **Gr**omov **Wa**sserstein **S**ubgraph Contrast**i**ve **L**earning (FOSSIL), is designed to be robust against variations in homophily levels within the graph dataset. The proposed method uses a decoupled scheme to process heterophilic data more effectively. Furthermore, FOSSIL leverages the FGWD to simultaneously encode the structural and feature characteristics of our subgraphs for calculating the CL loss. We conduct a comprehensive comparative analysis of FOSSIL against state-of-the-art models across various homophilic and heterophilic datasets for self-supervised node classification tasks. Our findings demonstrate that FOSSIL either outperforms existing methodologies or achieves competitive results, thereby showcasing its efficacy in graph SSL tasks. The main contributions of our paper can be summarized as follows:

- We introduce a novel approach, named FOSSIL, that effectively captures both feature and structural similarities simultaneously with the FGWD, addressing the limitations of previous OT-based SSL methods in GRL.

- We design a robust architecture capable of accommodating varying levels of homophily within graph datasets. By using a decoupled scheme for the GNN encoder, FOSSIL performs consistently well across homophilic and heterophilic datasets. This robustness enhances the applicability of the proposed method in real-world scenarios.

- We conduct an extensive empirical evaluation to validate the effectiveness of FOSSIL against state-of-the-art models in self-supervised node classification tasks across diverse datasets. We also thoroughly validate each design choice of FOSSIL through a series of ablation studies.

## 2 Related Work

### 2.1 Graph Neural Networks

GNNs are neural architectures designed for graphs (Wu et al., 2021). They usually extract information via message-passing mechanisms between nodes. This process can be divided into two steps: message aggregation from the neighborhood and updating of the node embedding (Gilmer et al., 2017). The key difference between state-of-the-art GNN architectures lies in the aggregation functions. For example, Graph Convolutional Networks (GCNs) (Kipf & Welling, 2017) perform a weighted average of the messages inside the 1-hop neighborhood, where weights are fixed and defined with respect to the node degrees. Graph Attention Networks (GATs) (Veličković et al., 2018) compute learnable attention weights thanks to interactions inside the 1-hop neighborhood. Nevertheless, most of the GNN architectures are built under the homophily as-

sumption. This assumption is useful in graph datasets where nodes have similar characteristics. Therefore, these GNNs have poor node classification performance in heterophilic graph datasets.

Previous research works have tried to address the heterophilic limitation of classical message-passing neural networks. For example, GPR-GNN (Chien et al., 2021) uses a polynomial graph filter with coefficients learned altogether with the linear projection layer used in GNNs. H2GCN (Zhu et al., 2020a) proposes a set of designs to improve the node classification performance of GNN on heterophilic graphs: feature-structure embedding separation, aggregating in higher-order neighborhoods, and a combination of intermediate representations (which is equivalent to adding skip connections between GNN layers). From a graph signal processing perspective (Isufi et al., 2024), GCNs are considered low-pass filters (NT & Maehara, 2019). Therefore, ACM (Luan et al., 2022) adaptively combines information extracted by a graph low-pass filter, a high-pass filter, and an MLP to address heterophilic networks.

In FOSSIL we adopt a decoupled encoder design similar to ACM (Luan et al., 2022), making our encoder agnostic to the input graph's level of homophily. Specifically, we use a GNN with a normalized adjacency matrix without self-loops and the identity, which implements a vanilla message-passing function and an MLP. These two representations are then adaptively combined with a fusion module. The message-passing module of our GNN is equivalent to the low-pass channel of ACM, while the MLP is the identity channel. Intuitively, we make our encoder more robust to different levels of homophily with the two representations.

## 2.2 Self-supervised Learning on Graphs

The key idea of SSL on graphs is to design and solve some pretext tasks that do not require label information. Depending on how the pretext task is defined, SSL methods for graphs can be divided into two categories: *predictive* and *contrastive* methods. Predictive methods aim to learn meaningful representations where the output is typically a perturbed version of the input graph. For example, BGRL (Thakoor et al., 2021) learns node representations by encoding two perturbed versions of the input graph with an online encoder and a target encoder. The online encoder is trained to predict the representation of the target encoder, and the target encoder is updated as an exponential moving average of the online encoder. BNLL (Liu et al., 2024) improves the BGRL model by introducing a few noisy-ground truth positive node pairs. Based on the homophily assumption that neighboring nodes often share the same label, they include additional candidate positive pairs to the objective of BGRL. Specifically, for each node, BNLL adds the cosine similarity of its online representation and the target representations of its neighbors weighted by its attention weights with its neighbors. VGAE (Kipf & Welling, 2016) uses a variational autoencoder approach in graphs to predict the same graph and feature as in the input. Finally, GPT-GNN (Hu et al., 2020b) performs an autoregressive reconstruction of the graph.

Contrastive methods, which are also the focus of this paper, have shown better performance than predictive methods for SSL on graphs in general. Such methods can be divided into three categories depending on how the data pairs are defined: node-to-node, graph-to-graph, and node-to-graph. For example, GRACE (Zhu et al., 2020b) generates two perturbed views of the original graph and performs contrastive learning between the nodes of these views. MUSE (Yuan et al., 2023) extracts node-wise semantic, contextual, and fused embeddings to perform node-to-node contrastive learning for each of these embeddings. Nevertheless, node-level contrastive learning approaches are suboptimal since they cannot easily capture global information about the structure of the graphs.

Regarding subgraph level contrast, DGI (Veličković et al., 2019) performs node-to-graph contrast by extracting the node embeddings of the original graph and a perturbed version, then it increases or decreases the agreement between the original or perturbed node embeddings using a readout function. Though spectral polynomial filters such as GPR-GNN (Chien et al., 2021) and ChebNetII (He et al., 2022) are more expressive than GCNs and can adapt to arbitrary homophily levels, they underperform GCNs when used as an encoder for traditional SSL methods. PolyGCL (Chen et al., 2024) tries to solve this problem by restricting the expressiveness of the polynomial filters from a spectral perspective to construct the low-pass and high-pass views and introduces a simple linear combination strategy to construct the optimization objective. Specifically, they follow a similar approach to DGI (Veličković et al., 2019) with the difference that they do it with both high/low-frequency embeddings extracted with high-pass and low-pass polynomial filters sharing

weights. Subg-Con (Jiao et al., 2020) follows up the work of DGI by doing the same contrast but at a sub-graph level. It starts by sampling a set of anchor nodes, followed by the extraction of subgraphs utilizing the personalized PageRank algorithm. Overall, the model aims to adjust the agreement level between an anchor node and the sampled subgraph for the positive and negative pairs. Methods such as DGI and Subg-Con, however, use a readout embedding to characterize graphs, which ignores the structure of the graph. GSC (Han et al., 2022) addresses this issue by performing subgraph-level contrast using WD and GWD from OT as the measure of subgraph similarity.

FOSSIL is a combined subgraph-level and node-level contrastive learning approach. Our model uses OT distance metrics to measure the dissimilarity between subgraphs while further incorporating node-level contrast to capture differences among nodes. While sharing common features, our method differs from GSC in three key aspects: (i) we jointly capture structural and feature information in the OT distance using the FGWD, (ii) we perform node-level contrast to emphasize the node's difference, and (iii) we have a specialized architecture that is robust to both homophilic and heterophilic datasets.

## 3 Preliminaries

### 3.1 Mathematical Notation

In this paper, matrices are represented with uppercase bold letters like $\mathbf{A}$, while lowercase bold letters like $\mathbf{x}$ denote vectors. The symbol $\langle \cdot, \cdot \rangle$ denotes the matrix scalar product associated with the Frobenius norm. For an element $x$ of a measured space, $\delta_x$ represents the Dirac measure at $x$. The vector $\mathbb{1}_n \in \mathbb{R}^n$ denotes the vector of all ones of dimension $n$. We denote the simplex histogram with $n$ bins as $\Sigma_n = \{\mathbf{h} \in \mathbb{R}_+^n \mid \sum_{i=1}^n \mathbf{h}_i = 1\}$. Let $\underline{\mathbf{L}}$ be a 4-way tensor and $\mathbf{B}$ be a matrix. The tensor-matrix multiplication between $\underline{\mathbf{L}}$ and $\mathbf{B}$ is denoted as the matrix $\underline{\mathbf{L}} \otimes \mathbf{B}$, where $(\underline{\mathbf{L}} \otimes \mathbf{B})_{i,j} = \sum_{k,l} \underline{\mathbf{L}}_{i,j,k,l} \mathbf{B}_{k,l}$. The element-wise (Hadamard) product operator between two matrices is denoted as $\odot$. The fraction $\frac{\mathbf{u}}{\mathbf{v}}$ denotes the element-wise division of vectors $\mathbf{u}$ and $\mathbf{v}$.

### 3.2 Graph Neural Networks

We consider node-attributed and undirected graphs, denoted as $\mathcal{G} = (\mathbf{A}, \mathbf{X})$, where $\mathbf{A} \in \mathbb{R}^{N \times N}$ represents the adjacency matrix, $\mathbf{X} \in \mathbb{R}^{N \times F}$ denotes the feature matrix, and $N$ and $F$ respectively denote the number of nodes and the dimensionality of each node feature vector. A GNN is a function $f : \mathbb{R}^{N \times N} \times \mathbb{R}^{N \times F} \rightarrow \mathbb{R}^{N \times D}$, where $\mathbf{H} = f(\mathbf{A}, \mathbf{X}) \in \mathbb{R}^{N \times D}$ represents the node embeddings matrix of $\mathcal{G}$, with each row $\mathbf{H}_i$ corresponding to the embedding of node $v_i$ (Wu et al., 2021). Additionally, for a vector $\mathbf{v} \in \mathbb{R}^N$, $\mathrm{diag}(\mathbf{v}) \in \mathbb{R}^{N \times N}$ denotes the diagonal matrix with diagonal elements from $\mathbf{v}$. Let $S \subseteq \{1, \ldots, N\}$ be a set of indices; $\mathbf{A}[S; S]$ represents the adjacency matrix of the subgraph of $\mathcal{G}$ restricted to nodes indexed by $S$, and $\mathbf{X}[S]$ denotes its corresponding feature matrix.

### 3.3 Fused Gromov-Wasserstein Distance

The main purpose of OT is to find the optimal way to shift objects from one location to another in a metric space. This is done by finding a minimum cost soft matching between two probability distributions, one on objects of each location. Let $\boldsymbol{\mu} \in \Sigma_n$ and $\boldsymbol{\nu} \in \Sigma_m$ be two probability distributions, respectively, over the objects of the first and the second location. The WD (Villani et al., 2009) uses a matrix $\mathbf{M} \in \mathbb{R}^{n \times m}$ of transportation costs (or pair-wise distances) between location 1 and 2 to find a soft matching that minimizes the overall transportation cost. However, locations 1 and 2 may not lie in the same space, preventing us from defining the transportation cost $\mathbf{M}$. GWD (Mémoli, 2011) solves this problem by using internal transportation cost matrices $\mathbf{C}_1$ and $\mathbf{C}_2$ in each space separately to find the soft matching. The purpose is to match objects that have similar structural behaviors inside their metric spaces.

The main downside of WD and GWD is that they fail to capture both structural and feature information (Vayer et al., 2018) when used to compare graphs. In fact, WD only uses node features to define its cost matrix $\mathbf{M}$, while GWD only uses each graph's adjacency matrix to define its internal costs matrices $\mathbf{C}_1$ and $\mathbf{C}_2$. To alleviate this issue, Vayer et al. (2018) proposed the Fused Gromov-Wasserstein Distance (FGWD).

The idea is to use a coefficient $\alpha \in [0,1]$ that trades off between WD and GWD. The FGWD between the two locations is defined as follows:

$$\text{FGWD}_{\alpha,\mathbf{M},\mathbf{C}_1,\mathbf{C}_2}(\boldsymbol{\mu},\boldsymbol{\nu}) = \min_{\mathbf{P}\in\Pi(\boldsymbol{\mu},\boldsymbol{\nu})} \langle \alpha\mathbf{M} + (1-\alpha)\underline{\mathbf{L}}(\mathbf{C}_1,\mathbf{C}_2) \otimes \mathbf{P}, \mathbf{P}\rangle, \tag{1}$$

where $\mathbf{P} \in \mathbb{R}^{n\times m}$ is a soft matching (or joint probability distribution) between objects of location 1 and 2, $\underline{\mathbf{L}}(\mathbf{C}_1,\mathbf{C}_2)_{i,j,k,l} = |\mathbf{C}_1[i;k] - \mathbf{C}_2[j;l]|^2$, and $\Pi(\boldsymbol{\mu},\boldsymbol{\nu})$ is the set of joint probability distributions with marginals $\boldsymbol{\mu}$ and $\boldsymbol{\nu}$. FGWD is a generalization of WD and GWD. We recover WD for $\alpha = 1$ and GWD for $\alpha = 0$.

### 3.4 Graph Contrastive Learning (GCL)

GCL (You et al., 2020; Xie et al., 2021) is built upon the Mutual Information Maximization principle (MIM) (Linsker, 1988). Given a graph $\mathcal{G}$ sampled from a graph distribution $\mathbb{P}_{\mathcal{G}}$, the goal is to learn an encoder $f(\cdot)$ by maximizing the mutual information between $\mathcal{G}$ and its encoded version $f(\mathcal{G})$. The encoder $f$ should be able to distinguish different graphs. Hence, we solve:

$$\max \text{MI}(\mathcal{G}; f(\mathcal{G})), \text{ where } \mathcal{G} \sim \mathbb{P}_{\mathcal{G}}. \tag{2}$$

For computational efficiency, in practice, we maximize an estimated lower bound $\widehat{\text{MI}}$ of the mutual information. These lower bounds are used to maximize the similarity between positive pairs and minimize the similarity between negative ones. Formally, given a graph $\mathcal{G} \sim \mathbb{P}_{\mathcal{G}}$, and a graph view generation operator $T(\mathcal{G})$ that creates different but semantically similar versions $t(\mathcal{G})$ of $\mathcal{G}$, the problem writes:

$$\max \widehat{\text{MI}}(f(\mathcal{G}), f(t(\mathcal{G}))), \text{ where } \mathcal{G} \sim \mathbb{P}_{\mathcal{G}} \text{ and } t(\mathcal{G}) \sim T(\mathcal{G}). \tag{3}$$

In this work, we use the Jensen-Shannon estimator (Nowozin et al., 2016) for our subgraph-level contrastive learning and the InfoNCE (van den Oord et al., 2018) for our node-level contrastive learning.

The learned representation $f(\mathcal{G})$ can then be transferred to a graph-level or node-level downstream task by plugging in a carefully designed decoder:

$$\min_{\boldsymbol{\theta}_{ds}} \mathcal{L}_{\text{downstream}}(\text{Dec}(f(\mathcal{G})), \mathcal{Y}), \tag{4}$$

where $\mathcal{Y}$, $\text{Dec}(\cdot)$, $\boldsymbol{\theta}_{ds}$, and $\mathcal{L}_{\text{downstream}}$ are respectively the labels, the decoder (or prediction head), the parameters, and the loss for the downstream task.

## 4 Fused Gromov-Wasserstein Subgraph Contrastive Learning (FOSSIL)

### 4.1 Overview

Figure 1 illustrates the pipeline of our proposed method, referred to as FOSSIL. FOSSIL comprises five primary stages: (i) *encoding*, (ii) view *generation*, (iii) *fusion* layer, (iv) subgraph *sampling*, and (v) *SSL loss* computation based on OT for subgraph-level plus a node-level contrastive loss. FOSSIL takes an input graph $\mathcal{G}$ and employs two neural networks with shared weights as encoders. The first neural network performs linear projection, while the second implements a GNN (Kipf & Welling, 2017), both equipped with the same learnable parameters $\mathbf{W}$. Subsequently, view generation follows a decoupled strategy similar to the encoder, utilizing a GAT (Veličković et al., 2018). The fusion layer combines the outputs from the encoder and generator using a linear combination. Finally, subgraphs are sampled at the encoder and generator levels to compute the subgraph-level CL loss using the FGWD as the distance metric, and a node-level contrastive loss is additionally computed.

### 4.2 Encoder

Several GNNs for self-supervised node classification are built under the homophily assumption, meaning that they assume the nodes in the neighborhood share similar labels (Han et al., 2022). As a consequence,

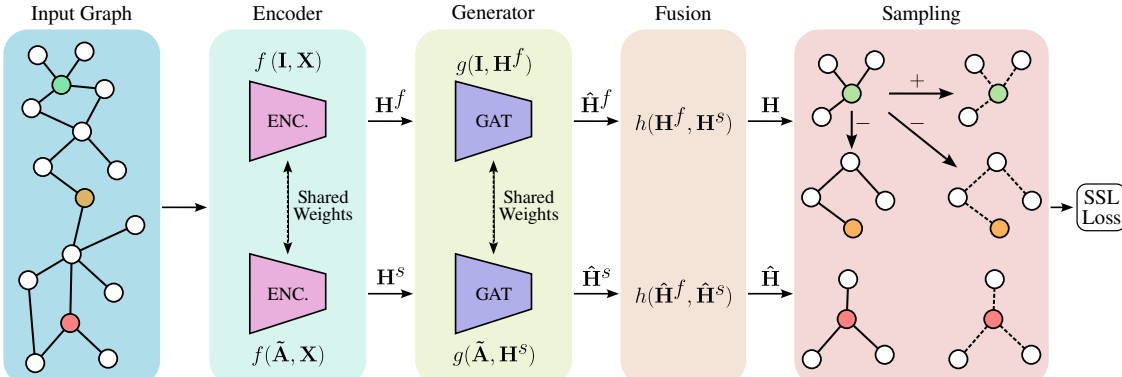

Figure 1: Pipeline of FOSSIL for self-supervised node classification tasks. The proposed method integrates encoding, view generation, fusion, subgraph sampling, and CL in attributed graphs. FOSSIL employs a shared-weight encoder and generator with GNNs to extract informative representations, subsequently fused for enhanced learning. Subgraph sampling facilitates subgraph-level CL loss computation based on the FGWD.

they typically perform badly when evaluated on heterophilic graph datasets. Several works (Zhu et al., 2020a; Hamilton et al., 2017; Defferrard et al., 2016; Abu-El-Haija et al., 2019; Chien et al., 2021) proposed new architectures to tackle this issue in the supervised domain. Surprisingly, these more powerful encoders underperform a simple GCN in current GCL methods (Chen et al., 2024). Based on these observations and following (Zhu et al., 2020a), we propose a decoupled encoding, where we use a GNN and an MLP with shared weights $\mathbf{W} \in \mathbb{R}^{F \times D}$. Nodes are encoded as follows:

$$\mathbf{H} = \sigma(\mathbf{IXW}) + \mathrm{diag}(\boldsymbol{\lambda})\sigma(\tilde{\mathbf{A}}\mathbf{XW}) = \mathbf{H}^f + \mathrm{diag}(\boldsymbol{\lambda})\mathbf{H}^s, \tag{5}$$

where $\sigma(\cdot)$ is a non-linearity, $\mathbf{I}$ is the identity matrix, $\tilde{\mathbf{A}} = \mathbf{D}^{-1/2}\mathbf{A}\mathbf{D}^{-1/2}$ is the normalized adjacency matrix, $\mathbf{D} = \mathrm{diag}(\mathbf{A}\mathbb{1}_N) \in \mathbb{R}^{N \times N}$ is the diagonal matrix of node degrees and $\boldsymbol{\lambda} \in [0, 1]^N$ a vector that dictates how the two terms are combined. We explain how to compute $\boldsymbol{\lambda}$ in more detail in Sec. 4.4.

In (5), we adaptively combine an MLP embedding, termed feature embedding, and a GNN embedding, termed structural embedding. Self-loops are not considered in the adjacency matrix, so the GNN only aggregates information from its neighbors. Intuitively, the decoupling strategy proves effective due to the presence of two distinct sources of information in an attributed graph: node features and graph structure. The relevance of each information source can vary depending on the downstream task at hand. For instance, in the context of a social network, individual user features often play a primary role in defining users, while friendship connections offer supplementary insights into their interactions. Conversely, in a citation network, it could be more crucial to examine the documents cited by an article to predict its category effectively. In essence, on homophilic graphs, traditional GNN encoding typically outperforms MLPs, whereas on heterophilic graphs, MLPs tend to perform better (Zhu et al., 2020a). The decoupled approach enables us to better adapt the SSL model to the homophily level of the graph.

## 4.3 Generator

Generating positive pairs in SSL for graphs is still an open problem of research. A common approach here involves perturbing the original graph to preserve the underlying semantics (You et al., 2020; Qiu et al., 2020). However, determining these perturbations for graphs is not straightforward. State-of-the-art methods usually apply hand-crafted perturbations like random node feature masking and random edge dropping. Such perturbations lack control and can degrade the intrinsic properties of the original graph—hence increasing the risk of false positives (Wang et al., 2022). Moreover, the performance of these hand-crafted perturbations heavily depends on the characteristics of the particular graph dataset. Therefore, an additional step is required to manually identify the appropriate perturbation method tailored to each dataset.

In FOSSIL, we adopt an end-to-end perturbation strategy, meaning that we learn the correct perturbation from the data. We use a similar strategy as in the encoder to separately generate node feature perturbations and graph structure perturbations. Given the feature embeddings $\mathbf{H}^f$ and structure embeddings $\mathbf{H}^s$ from the encoder, we generate perturbations as $\hat{\mathbf{H}}^f = g(\mathbf{I}, \mathbf{H}^f)$, $\hat{\mathbf{H}}^s = g(\tilde{\mathbf{A}}, \mathbf{H}^s)$, where $g : \mathbb{R}^{N \times N} \times \mathbb{R}^{N \times F} \to \mathbb{R}^{N \times D}$ corresponds to a GAT model (Veličković et al., 2018). Intuitively, we leverage the attention weights of the GAT to adaptively generate perturbations. At the feature level, this is equivalent to applying adaptive scaling to the node features, while at the graph structure level, this is equivalent to adaptively dropping some edges.

## 4.4   Fusion Layer

We combine the feature and structure information from the encoder and generator to obtain the final node embeddings. To this end, we apply adaptive node-wise fusion layers with shared weights $\boldsymbol{\theta}$. This stems from the fact that nodes do not necessarily require the same amount of structural information. At the encoder level, for each node $v_i$ we combine the semantic and structural information as follows:

$$\boldsymbol{\lambda}_i = \psi(\mathbf{H}_i^f, \mathbf{H}_i^s, (\mathbf{A}\mathbb{1}_N)_i; \boldsymbol{\theta}), \tag{6}$$

$$\mathbf{H}_i = h(\mathbf{H}_i^f, \mathbf{H}_i^s) = \mathbf{H}_i^f + \boldsymbol{\lambda}_i \mathbf{H}_i^s, \tag{7}$$

where $\psi : \mathbb{R}^D \times \mathbb{R}^D \times \mathbb{R} \to \mathbb{R}$ is an MLP (with sigmoid as the last activation function) computing how much structural information each node needs, and $\boldsymbol{\lambda} \in [0,1]^N$ is the vector of the fusion coefficients of the graph nodes. Similarly, for the generator, we have the following:

$$\hat{\boldsymbol{\lambda}}_i = \psi(\hat{\mathbf{H}}_i^f, \hat{\mathbf{H}}_i^s, (\mathbf{A}\mathbb{1}_N)_i; \theta), \tag{8}$$

$$\hat{\mathbf{H}}_i = h(\hat{\mathbf{H}}_i^f, \hat{\mathbf{H}}_i^s) = \hat{\mathbf{H}}_i^f + \hat{\boldsymbol{\lambda}}_i \hat{\mathbf{H}}_i^s. \tag{9}$$

Following (Yuan et al., 2023), we incorporate the degrees of the nodes $\mathbf{A}\mathbb{1}_N$ into the fusion module. The rationale behind this choice is to leverage the node degree centralities to account for the connectivity information of each node inside the graph. Therefore, we help function $\psi(\cdot)$ in computing the fusion coefficients $\boldsymbol{\lambda}$ that are more sensitive to the structure of the graph.

## 4.5   Subgraph Sampling

Using the original node embeddings $\mathbf{H}$ and the perturbed node embeddings $\hat{\mathbf{H}}$, we sample the subgraphs that will be subsequently used as contrastive samples. To this end, we first sample without replacement a set of anchor-node indices $S \subseteq \{1, \dots, N\}$. From each anchor node $v_i \in S$, we sample a set of node indices $S_i$ of size $k$ (including $v_i$) using breadth-first sampling in the 2-hop neighborhood of $v_i$. If we do not get the required size $k$, then the subgraph is not included in the subgraph-level contrast. Therefore, we create two subgraphs using each $S_i$. The first subgraph is given by $\mathcal{G}_i^s = (\mathbf{A}[S_i; S_i], \mathbf{H}[S_i])$. The second subgraph is given by $\hat{\mathcal{G}}_i^s = (\hat{\mathbf{A}}[S_i; S_i], \hat{\mathbf{H}}[S_i])$, where $\hat{\mathbf{A}}_{i,j} = s(\hat{\mathbf{h}}_i, \hat{\mathbf{h}}_j) = \frac{\hat{\mathbf{h}}_{s_i}^\top \hat{\mathbf{h}}_{s_j}}{\|\hat{\mathbf{h}}_{s_i}\|_2 \|\hat{\mathbf{h}}_{s_j}\|_2}$. These subgraphs are used in our CL loss function using the FGWD. A perturbed subgraph corresponds to the original subgraph contextualized by the GAT we are using for adaptive graph perturbation. The intuition is that this contextualized subgraph further captures the similarities between nodes inside the original subgraph, hence, it should be similar to the original subgraph while adding additional context information to the nodes. In this work, for simplicity, we choose to randomly sample subgraphs. Some studies have proposed more adaptive and data-driven subgraph sampling techniques, which present an interesting direction for future work in subgraph contrastive learning (Niepert et al., 2021; Bevilacqua et al., 2024; Kong et al., 2023). This subgraph sampling approach breaks the equivariance of our encoder with respect to the contrastive loss. However, this does not negatively impact the performance of FOSSIL in the datasets we use (please see Sec. 5).

## 4.6   Self-supervised Loss Function

The SSL loss function in FOSSIL is given by two terms: a contrastive loss $\mathcal{L}_{\text{contrast}}$, and a regularization term $\mathcal{L}_\theta$ to guide the learning of the fusion layer. The contrastive loss is at the same time divided into an

OT loss $\mathcal{L}_{\mathrm{ot}}$ and a node-level contrastive loss $\mathcal{L}_{\mathrm{node}}$, so that $\mathcal{L}_{\mathrm{contrast}} = \mathcal{L}_{\mathrm{ot}} + \mathcal{L}_{\mathrm{node}}$. Our final loss is thus given by $\mathcal{L} = \mathcal{L}_{\mathrm{contrast}} + \mathcal{L}_{\theta}$.

### 4.6.1 OT distance-based subgraph-level contrastive loss $\mathcal{L}_{\mathbf{ot}}$

In FOSSIL, we consider graphs as measured spaces by endowing them with probability distributions over their nodes. The probability mass on a given node represents the node's relative importance in the graph. With no prior knowledge of the graphs, we assume a uniform probability distribution over its nodes.

Let $\mathcal{G}_1^s = (\mathbf{A}_1^s, \mathbf{H}_1^s)$ and $\mathcal{G}_2^s = (\mathbf{A}_2^s, \mathbf{H}_2^s)$ be two subgraphs. We measure the difference between $\mathcal{G}_1^s$ and $\mathcal{G}_2^s$ using the FGWD as follows:

$$D_{\mathrm{fgw}}(\mathcal{G}_1^s, \mathcal{G}_2^s) = \mathrm{FGWD}_{\alpha, \mathbf{C}^{1,2}, \mathbf{C}^1, \mathbf{C}^2}(\boldsymbol{\mu}, \boldsymbol{\nu}), \tag{10}$$

where $\mu$ is the uniform distribution over nodes of $\mathcal{G}_1^s$, $\nu$ is the uniform distribution on the nodes of $\mathcal{G}_2^s$, $\mathbf{C}^1 = \exp(-\mathbf{A}_1^s/\tau)$, $\mathbf{C}^2 = \exp(-\mathbf{A}_2^s/\tau)$ and $\mathbf{C}^{1,2} = \exp(-\mathbf{H}_1^s(\mathbf{H}_2^s)^{\top}/\tau)$ is the matrix of pairwise transportation costs between their nodes (the exponentials on matrices are applied element-wise). We use FGWD to jointly exploit both features and structure information inside the two graphs to compute their distances. In this work, we solve the optimization problem associated with FGWD using the Bregman Alternated Projected Gradient (BAPG) method (Li et al., 2023) given in Alg. 1, App. A. Our subgraph-level contrastive loss can be thus written as:

$$\begin{aligned}
\mathcal{L}_{\mathrm{ot}} = -\frac{1}{|S|(M+1)} \sum_{i \in S} &\Big[ \log\Big( \sigma(\exp(-D_{\mathrm{fgw}}(\mathcal{G}_i^s, \hat{\mathcal{G}}_i^s)/\tau)) \Big) \\
&+ \sum_{j=1}^{M} \log\big( 1 - \sigma(\exp(-D_{\mathrm{fgw}}(\mathcal{G}_i^s, \mathcal{G}_i^{sn_j})/\tau))) \big) \Big].
\end{aligned} \tag{11}$$

where $S$ is the set of sampled anchor-node indices defined in Sec. 4.5, $\mathcal{G}_i^s$ is a sampled subgraph, and $\mathcal{G}_i^{sn_j}$ is its $j$-th negative view used in the contrastive loss. For each node, we define $M$ negative views in (11). In practice, we set $M = 2$ and define $\mathcal{G}_i^{sn_1} = \mathcal{G}_j^s$ and $\mathcal{G}_i^{sn_2} = \hat{\mathcal{G}}_j^s$ with $j$ randomly picked in $\{j \in S : j \neq i\}$. In the contrastive loss $\mathcal{L}_{\mathrm{ot}}$, $\mathcal{L}_{\mathrm{align}}$ aims to increase the agreement between positive pairs while $\mathcal{L}_{\mathrm{reg}}$ aims to decrease the agreement between negative pairs.

### 4.6.2 Node-level contrastive loss $\mathcal{L}_{\mathbf{node}}$

We also incorporate node-level contrastive loss. This improves our downstream node classification performance, as demonstrated later in our experiments. The node-level loss writes:

$$l(\mathbf{h}_i, \hat{\mathbf{h}}_i) = -\log \frac{\exp(s(\mathbf{h}_i, \hat{\mathbf{h}}_i)/\tau)}{\sum_{j \neq i} \exp(s(\mathbf{h}_i, \mathbf{h}_j)/\tau) + \sum_{j=1}^{N} \exp(s(\mathbf{h}_i, \hat{\mathbf{h}}_j)/\tau)} \tag{12}$$

$$\mathcal{L}_{\mathrm{node}} = \frac{1}{2N} \sum_{i=1}^{N} \Big( l(\mathbf{h}_i, \hat{\mathbf{h}}_i) + l(\hat{\mathbf{h}}_i, \mathbf{h}_i) \Big). \tag{13}$$

One may notice that (12) presents a memory bottleneck coming from the contrastive loss at the node level $\mathcal{L}_{\mathrm{node}}$. This loss requires the storage of a $N \times N$ matrix for the pairwise cosine similarities between all pairs of nodes within the graph, hence $\mathcal{O}(N^2)$ of memory complexity. This limits the scalability of FOSSIL to large-scale graphs. To avoid this issue, we propose a FOSSILv2 model that considers only the union of nodes within the sampled subgraphs when computing the contrastive loss at the node level. Hence, the alternative node-level contrastive loss becomes:

$$\mathcal{L}_{\mathrm{node-v2}} = \frac{1}{2\left|\bigcup_{j \in S} S_j\right|} \sum_{i \in \bigcup_{j \in S} S_j} \Big( l(\mathbf{h}_i, \hat{\mathbf{h}}_i) + l(\hat{\mathbf{h}}_i, \mathbf{h}_i) \Big). \tag{14}$$

With this expression, we have a memory complexity of $\mathcal{O}((|S|k)^2 D)$, which scales constantly with respect to $N$ for FOSSILv2.

### 4.6.3 Embedding fusion loss $\mathcal{L}_\theta$

The embedding fusion loss guides the fusion layer learning behavior. The main motivation for this loss is to prevent the structure and feature embeddings from the encoder from being too similar. To this end, when the structure and feature embedding are very similar, we assume that we should emphasize more the feature embedding. Therefore, we promote low values in the vector $\boldsymbol{\lambda}$ for the concerned nodes. We also couple the $\mathcal{L}_\theta$ and the OT loss through the parameter $\alpha$. Notice that $1 - \alpha$ in the definition of the FGWD in (1) represents the relative weight of structure similarity when comparing subgraphs with OT. Therefore, we promote the values of $\boldsymbol{\lambda}$ to align with $1 - \alpha$ on average so that $\boldsymbol{\lambda}$ aligns with the OT distances. Additionally, we constrain the variance of $\boldsymbol{\lambda}$ with $L_2$ regularization to put more control on the way we combine embeddings. Our fusion layer loss function is thus given as:

$$\mathcal{L}_\theta = \sum_{i=1}^{N} \boldsymbol{\lambda}_i s(\mathbf{h}_i^s, \mathbf{h}_i^c) + \beta_1 \|\boldsymbol{\lambda}\|_2 + \beta_2 \left| \frac{1}{N} \sum_{i=1}^{N} \boldsymbol{\lambda}_i - (1 - \alpha) \right|, \tag{15}$$

where $\beta_1$ and $\beta_2$ are regularization parameters.

## 4.7 Computational Complexity of the Framework

### 4.7.1 Time complexity

Here, we analyze the computational complexity of our subgraph-level contrastive loss in (10). To compute FGWD, we use the BAPG method described in Alg. 1 of App. A, which is a single-loop method for solving the non-convex quadratic optimization problem of FGWD. The tensor product $\underline{\mathbf{L}}(\mathbf{C}_1, \mathbf{C}_2) \otimes \mathbf{P}$ can be computed with complexity $\mathcal{O}(n^2 m + nm^2)$. So the gradients computed at lines 4 and 8 of Alg. 1 can be computed with complexity $\mathcal{O}(n^2 m + nm^2)$. The other operations of Alg. 1 have complexity $\mathcal{O}(nm)$. Hence, the time complexity of BAPG is $\mathcal{O}(T(n^2 m + nm^2))$ overall, where $n$ and $m$ are, respectively, the order of the first and second graph, and $T$ is the number of iterations that have been performed. Now we need to compute the cost matrices $\mathbf{M}$, $\mathbf{C}_1$, and $\mathbf{C}_2$ that are inputs of the BAPG algorithm. Those are computed with an overall complexity $\mathcal{O}(nmD)$, where $D$ is the dimension of the embedding vectors. Finally solving the optimization problem in (1) with Alg. 1 has complexity $\mathcal{O}(T(n^2 m + nm^2) + nmD)$.

At each iteration of our method, we sample $|S|$ subgraphs, each with $k$ nodes (so we are in the case where $n = m = k$). Then, for each subgraph, we compute its FWGDs with its positive view and its $M$ negative views, which costs $\mathcal{O}((M + 1)(Tk^3 + k^2 D))$ in time. Hence, solving all the optimization problems costs $\mathcal{O}(|S|(M + 1)(Tk^3 + k^2 D))$ in time. Finally, we need to compute the contrastive loss with the FGWDs, which is done in $\mathcal{O}(|S|(M+1))$. Overall the time complexity of computing $\mathcal{O}(|S|(M + 1)(Tk^3 + k^2 D))$.

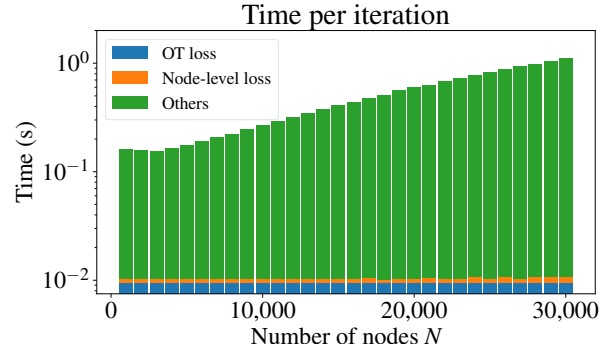

Figure 2: Time per iteration of the different components of FOSSIL in a GPU A40 48G.

It is worth noting that this time complexity only depends on the hyperparameters of our method. In particular, the cubic term $k^3$ is not a real concern in practice, as $k$ is chosen such that $k \ll N$, with $N$ the number of nodes in the graph. Therefore, the complexity of computing $\mathcal{L}_{\mathrm{ot}}$ does not have scalability issues with the size of the input graph as it remains constant w.r.t. $N$.

In our framework, we also compute a node-level contrastive loss $\mathcal{L}_{\mathrm{node}}$. This loss is computed with complexity $\mathcal{O}(N^2 D)$ theoretically. However, in practice, as computing the pair-wise cosine similarities between nodes inside the graph is parallelized thanks to matrix multiplication on the GPU, the complexity is roughly constant depending on the number of cores and memory available on the GPU.

To experimentally validate this theoretical analysis, we evaluate the time required for computing each loss with synthetic graphs generated with the contextual stochastic block model (cSBM) (Chien et al., 2021).

Table 1: Statistics of the datasets used in this work.

|  | Cora | CiteSeer | PubMed | CoAuthor-CS | Actor | Chameleon | Squirrel |
|---|---|---|---|---|---|---|---|
| $H(G)$ | 0.81 | 0.74 | 0.80 | 0.81 | 0.22 | 0.24 | 0.22 |
| Nodes | $2,708$ | $3,327$ | $19,717$ | $18,333$ | $7,600$ | $2,277$ | $5,201$ |
| Edges | $10,556$ | $9,104$ | $88,648$ | $163,788$ | $30,019$ | $36,101$ | $217,073$ |
| Features | $1,433$ | $3,703$ | $500$ | $6,805$ | $932$ | $2,325$ | $2,089$ |
| Classes | 7 | 6 | 3 | 15 | 5 | 5 | 5 |

We vary the number of nodes $N$ from $1,000$ to $30,000$, and for each value of $N$, we measure the times on 5 synthetic graphs with arbitrary homophily levels. We fix $k = 12$, $D = 512$, $|S| = 300$, and $M = 2$ for this experiment. Fig. 2 shows that the time required to compute the OT loss and node-level contrastive losses remains roughly constant for any value of $N$. Overall, in Fig. 2, we observe that our method scales reasonably well with the size of the graph. The empirical time complexity depicted by the green bars does not significantly increase from a graph with $1,000$ nodes to one with $30,000$ nodes.

### 4.7.2 Memory complexity

Computing the FGWD with Alg. 1 requires storing all the cost matrices and the intermediate matrices used inside the algorithm. Hence, this entails a memory complexity of $\mathcal{O}(n^2 + m^2 + nm)$ for two graphs of order $n$ and $m$. In our case, $n = m = k$, and we compute the FGWD of all the $|S|(M+1)$ subgraph pairs together inside the same batch. Therefore, we have a memory complexity of $\mathcal{O}(k^2|S|(M+1))$ for $\mathcal{L}_{ot}$. The memory complexity of the optimal transport component of FOSSIL is thus constant w.r.t. the size of the graph $N$.

## 5 Experimental Framework and Results

We perform a set of experiments to compare our model with several approaches in the literature. We compare FOSSIL with nine state-of-the-art methods including: GSC (Han et al., 2022), DGI (Veličković et al., 2019), DSSL (Xiao et al., 2022), BGRL (Thakoor et al., 2021), GRACE (Zhu et al., 2020b), MUSE (Yuan et al., 2023), Subg-Con (Jiao et al., 2020), PolyGCL (Chen et al., 2024), and BNLL (Liu et al., 2024). In addition, we compare FOSSIL with GSC using as the encoder the Chebyshev graph convolution (Defferrard et al., 2016) (GSC+Cheb) instead of the regular GCN to make the GSC framework more robust to heterophilic datasets. We evaluate all methods in four homophilic datasets Cora (McCallum et al., 2000), CiteSeer (Sen et al., 2008), PubMed (Namata et al., 2012), and CoAuthor-CS (Shchur et al., 2018); and in three heterophilic datasets Actor (Tang et al., 2009), Chameleon, and Squirrel (Rozemberczki et al., 2021). We perform additional experiments in large-scale datasets in App. B. Table 1 shows the statistics of the datasets tested in this work, where $H(G)$ is the homophily of the graph as defined in (Pei et al., 2020). Finally, we also run a set of ablation studies to analyze different aspects of FOSSIL.

We use a standard downstream logistic regression accuracy with the self-supervised node embeddings to evaluate all compared methods. We split the data into a development set (80%) and a test set (20%) once, ensuring that the test set is not used during the hyperparameter optimization process. Therefore, we split the development set into training and validation sets for hyperparameter tuning. Following (Klicpera et al., 2019; Giraldo et al., 2023), we take 20 nodes per class for the training set and the rest into the validation set for Cora, CiteSeer, and PubMed. For the other datasets, the training set gets 60% of the nodes, and the validation set gets the other 40% in the development set. Finally, we test each method using 100 test seeds to randomly split the dataset into the train and validation, while keeping the original test set. We train and evaluate the encoder on those splits and report the average performances accompanied by 95% confidence intervals calculated by bootstrapping with $1,000$ samples. Unlike other works in SSL for graphs, we do not perform model selection during SSL training by looking at the downstream test/validation performance. Finally, we take the encoder at the end of the SSL training to extract the node embeddings for the downstream evaluation. All experiments are conducted on A40 48GB and P100 16GB GPUs.

Table 2: Test accuracy on the node classification task for different baseline models and FOSSIL. The best results are highlighted in **bold**, while the second best-performing methods are underlined.

| Method | Cora | CiteSeer | PubMed | CoAuthor CS | Actor | Chameleon | Squirrel |
|---|---|---|---|---|---|---|---|
| FS-GCN | $80.04_{\pm0.26}$ | $69.05_{\pm0.26}$ | $77.33_{\pm0.41}$ | $92.50_{\pm0.02}$ | $29.69_{\pm0.14}$ | $45.48_{\pm0.25}$ | $30.26_{\pm0.18}$ |
| FS-MLP | $57.28_{\pm0.39}$ | $56.37_{\pm0.37}$ | $69.20_{\pm0.43}$ | $92.05_{\pm0.06}$ | $30.08_{\pm0.20}$ | $48.68_{\pm0.33}$ | $37.14_{\pm0.16}$ |
| GSC | $79.29_{\pm0.20}$ | $54.56_{\pm0.74}$ | $76.98_{\pm0.50}$ | $90.16_{\pm0.06}$ | $28.43_{\pm0.16}$ | $42.90_{\pm0.40}$ | $28.98_{\pm0.22}$ |
| GSC+Cheb | $76.84_{\pm0.26}$ | $68.11_{\pm0.26}$ | $75.75_{\pm0.45}$ | $89.20_{\pm0.14}$ | $32.46_{\pm0.13}$ | $43.99_{\pm0.34}$ | $35.03_{\pm0.24}$ |
| MUSE | $62.52_{\pm0.38}$ | $58.05_{\pm0.42}$ | $79.08_{\pm0.31}$ | $54.32_{\pm1.76}$ | $\mathbf{36.61}_{\pm0.15}$ | $52.11_{\pm0.50}$ | $33.57_{\pm0.17}$ |
| DGI | $79.90_{\pm0.23}$ | $\underline{70.22}_{\pm0.32}$ | $77.80_{\pm0.47}$ | $87.38_{\pm0.06}$ | $29.11_{\pm0.14}$ | $36.24_{\pm0.24}$ | $29.58_{\pm0.16}$ |
| DSSL | $72.39_{\pm0.56}$ | $54.16_{\pm0.64}$ | $69.98_{\pm0.69}$ | $87.73_{\pm0.08}$ | $26.60_{\pm0.14}$ | $48.38_{\pm0.35}$ | $34.99_{\pm0.21}$ |
| BGRL | $62.95_{\pm0.87}$ | $38.02_{\pm1.46}$ | $72.93_{\pm0.44}$ | $91.81_{\pm0.22}$ | $27.93_{\pm0.17}$ | $36.90_{\pm0.49}$ | $28.91_{\pm0.25}$ |
| GRACE | $\underline{80.86}_{\pm0.29}$ | $65.86_{\pm0.40}$ | $\mathbf{79.84}_{\pm0.33}$ | $93.01_{\pm0.03}$ | $28.91_{\pm0.18}$ | $43.87_{\pm0.34}$ | $30.60_{\pm0.22}$ |
| Subg-Con | $44.32_{\pm0.02}$ | $30.07_{\pm0.91}$ | $67.35_{\pm0.85}$ | $82.29_{\pm0.54}$ | $27.60_{\pm0.18}$ | $45.60_{\pm0.44}$ | $\mathbf{37.16}_{\pm0.43}$ |
| PolyGCL | $\mathbf{80.95}_{\pm0.26}$ | $\mathbf{71.38}_{\pm0.21}$ | $76.39_{\pm0.41}$ | OOM | $26.52_{\pm0.07}$ | $32.75_{\pm0.19}$ | $34.25_{\pm0.13}$ |
| BNLL | $66.46_{\pm0.83}$ | $41.57_{\pm1.17}$ | $70.45_{\pm0.92}$ | $93.60_{\pm0.03}$ | $25.33_{\pm0.34}$ | $50.18_{\pm0.90}$ | $36.63_{\pm0.50}$ |
| FOSSIL | $80.02_{\pm0.26}$ | $68.24_{\pm0.34}$ | $\underline{79.76}_{\pm0.48}$ | $\underline{94.49}_{\pm0.04}$ | $\underline{35.61}_{\pm0.15}$ | $\underline{53.03}_{\pm0.33}$ | $\underline{37.13}_{\pm0.24}$ |
| FOSSILv2 | $78.94_{\pm0.26}$ | $68.16_{\pm0.23}$ | $79.09_{\pm0.39}$ | $\mathbf{95.04}_{\pm0.03}$ | $35.41_{\pm0.18}$ | $\mathbf{53.14}_{\pm1.24}$ | $34.79_{\pm0.21}$ |

## 5.1 Implementation Details

We implement all methods using PyTorch (Ketkar, 2017) and PyG (Fey & Lenssen, 2019). Our encoder is a two-layer GCN with hidden dimension $1,024$ and output dimension $512$, and a PReLU (He et al., 2015) activation function between them. In all experiments, our method is trained for 300 epochs with the Adam (Kingma & Ba, 2015) optimizer. We optimize the other hyperparameters with the framework Optuna (Akiba et al., 2019). The search spaces for hyperparameters are defined as follows: 1) learning rate $lr \in \{10^{-4}, 5 \times 10^{-4}, 10^{-3}, 5 \times 10^{-3}, 10^{-2}\}$; 2) learning rate of fusion module $lr_f \in \{10^{-4}, 5 \times 10^{-4}, 10^{-3}, 5 \times 10^{-3}, 10^{-2}\}$; 3) $\alpha \in \{[0 : 0.1 : 1]\}$; 4) the FGWD regularizer $\beta \in \{10^{-3}, 5 \times 10^{-3}, 10^{-2}, 5 \times 10^{-2}, 10^{-1}, 5 \times 10^{-1}, 1, 1.5, 2\}$; 5) the number of nodes in each sampled subgraph $k \in [10, 30]$; 6) the temperature parameter $\tau \in \{0.2, 0.5, 0.8, 1.0, 1.5, 2.0, 2.5, 3.0\}$; 7) the GNN dropout parameter $p \in \{0.1, 0.2, 0.3, 0.4\}$, and 8) the fusion MLP dropout $p_f \in \{0.1, 0.2, 0.3, 0.4\}$. The value of $\beta_2$ is fixed to 1. We tune the hyperparameters by performing 100 trials in the development set. Finally, for each trial, we sample the values from the search space using the Tree-structured Parzen estimator (Watanabe, 2023). We study the sensibility of some important hyperparameters of FOSSIL in App. C. The source code of FOSSIL is publicly available [1].

## 5.2 Results and Discussion

Table 2 shows the results of the comparison of FOSSIL against previous state-of-the-art models for self-supervised node classification. We evaluate all methods with our setting to have a fair comparison. Our encoder is a decoupled GCN without self-loops; thus, we also evaluate GCN in a fully supervised setting, which we refer to as FS-GCN. We also evaluate a fully-supervised MLP (FS-MLP) for reference.

In Table 2, we observe that FOSSIL outperforms FS-MLP and FS-GCN, suggesting that pre-trained models using self-supervision and then fine-tuned on a downstream task can outperform end-to-end supervised models. In general, FOSSIL ranks either as the best or second best method among the baseline models used across both homophilic and heterophilic datasets. We also observe that FOSSIL is better than MUSE and GRACE on average, showing that node-level only contrast is suboptimal compared to including subgraph-level contrast for node classification.

It is also important to highlight that FOSSIL outperforms DGI and Subg-Con, which use a readout function to characterize a graph. This suggests that a simple readout function does not properly capture the structure of the graph. Besides, these methods perform node-graph contrast. Maximizing the agreement of an anchor node and its sampled subgraph is not optimal for node classification since this subgraph can contain a

---

[1] https://github.com/sangaram/FOSSIL

Table 3: Impact of subgraph similarity metric in node classification.

| Cosim | WD | GWD | FGWD | $\mathcal{L}_{\text{node}}$ | Cora | CiteSeer | PubMed | CoAuthor CS | Actor | Chameleon | Squirrel |
|---|---|---|---|---|---|---|---|---|---|---|---|
| ✓ | ✗ | ✗ | ✗ | ✗ | $66.04_{\pm0.54}$ | $61.32_{\pm0.44}$ | $70.13_{\pm0.63}$ | $84.33_{\pm0.28}$ | $34.27_{\pm0.17}$ | $42.43_{\pm0.29}$ | $33.35_{\pm0.29}$ |
| ✗ | ✓ | ✗ | ✗ | ✗ | $61.82_{\pm0.44}$ | $59.06_{\pm0.39}$ | $71.94_{\pm0.68}$ | $91.22_{\pm1.83}$ | $35.64_{\pm0.18}$ | $44.20_{\pm0.36}$ | $33.09_{\pm0.26}$ |
| ✗ | ✗ | ✓ | ✗ | ✗ | $60.73_{\pm0.89}$ | $50.88_{\pm0.50}$ | $68.56_{\pm0.76}$ | $90.98_{\pm0.10}$ | $33.57_{\pm1.45}$ | $52.42_{\pm1.05}$ | $31.07_{\pm0.82}$ |
| ✗ | ✗ | ✗ | ✓ | ✗ | $75.19_{\pm0.42}$ | $65.22_{\pm0.29}$ | $77.59_{\pm0.43}$ | $\underline{94.61}_{\pm0.04}$ | $\mathbf{35.79}_{\pm0.18}$ | $48.95_{\pm0.60}$ | $33.61_{\pm0.22}$ |
| ✓ | ✗ | ✗ | ✗ | ✓ | $78.00_{\pm0.35}$ | $\underline{69.14}_{\pm0.30}$ | $\underline{78.58}_{\pm0.43}$ | $94.80_{\pm0.03}$ | $35.11_{\pm0.17}$ | $\underline{55.42}_{\pm1.08}$ | $35.66_{\pm0.41}$ |
| ✗ | ✓ | ✗ | ✗ | ✓ | $78.23_{\pm0.29}$ | $\mathbf{69.95}_{\pm0.23}$ | $74.56_{\pm0.56}$ | $94.16_{\pm0.03}$ | $31.22_{\pm0.15}$ | $\mathbf{60.37}_{\pm0.49}$ | $35.25_{\pm0.25}$ |
| ✗ | ✗ | ✓ | ✗ | ✓ | $\underline{79.11}_{\pm0.31}$ | $67.72_{\pm0.29}$ | $77.30_{\pm0.41}$ | $94.01_{\pm0.04}$ | $34.29_{\pm1.13}$ | $50.51_{\pm0.36}$ | $35.25_{\pm0.20}$ |
| ✗ | ✗ | ✗ | ✓ | ✓ | $\mathbf{80.02}_{\pm0.26}$ | $68.24_{\pm0.34}$ | $\mathbf{79.76}_{\pm0.48}$ | $94.49_{\pm0.04}$ | $\underline{35.61}_{\pm0.15}$ | $53.03_{\pm0.33}$ | $\underline{37.13}_{\pm0.24}$ |

majority of dissimilar nodes (particularly in heterophilic graphs)—hence, its readout representation can be a false positive. This can be observed by the performance of DGI on heterophilic graphs.

FOSSIL works better than the previous OT-based method GSC, showing that a joint feature-structure subgraph similarity metric together with a decoupled embedding extraction benefits from the structural patterns inside the graph. Therefore, FOSSIL induces more relevant perturbations. Resulting in a more accurate contrastive training. Interestingly, FOSSILv2 has competitive performance in all compared datasets while being more scalable than many other methods of the literature. Therefore, FOSSILv2 offers a good compromise between performance and scalability. This suggests that our proposed model is a promising direction when scalability is a key factor.

## 5.3 Ablation Studies

We validate the effectiveness of each component of FOSSIL through a series of ablation studies. First, we validate the use of the FGWD that captures both feature and graph structure similarities between subgraphs. Notably, we demonstrate that adding the node-level contrastive loss consistently improves our downstream node-classification performance. Then, we validate the design of our encoder by testing other alternatives. We also explore other strategies to generate graph perturbations within our framework. Finally, we validate the usage of using node connectivity information for the fusion module $\psi(\cdot)$ to assist the computation of $\boldsymbol{\lambda}$.

### 5.3.1 Impact of the subgraph similarity metric

We replace the FGWD with other subgraph similarity metrics to evaluate its effectiveness. More precisely, we use other OT distances like WD and GWD. We also use a simple average pooling of the subgraphs' node features followed by a cosine similarity (referred to as Cosim). To properly assess the effect of every similarity function, we evaluate their performance with and without adding the node-level contrastive loss $\mathcal{L}_{\text{node}}$. Table 3 shows the results of this ablation study in all datasets. We first notice that node features are more important than pure graph connectivity for the node classification tasks. Indeed, GWD, which relies solely on subgraph structure similarity, performs worse than WD and Cosim. Secondly, we observe that WD performs better than Cosim. This suggests that OT distances are better for self-supervised learning in estimating graph similarities than a Readout function followed by a cosine similarity. Nevertheless, both WD and Cosim ignore the underlying graph structure. Interestingly, we observe the superior performance of using the FGWD that jointly captures the feature and structure similarities between subgraphs. This aligns with the observations of (Vayer et al., 2018). Finally, we notice that adding a node-level contrastive loss consistently improves our node classification results.

### 5.3.2 Impact of encoder design

We analyze the effectiveness of our encoder by replacing it with a vanilla GCN and a GCNII (Chen et al., 2020a) architecture. As shown in Table 4, the decoupled approach outperforms the standard GCN on the heterophilic datasets. Moreover, GCNII also underperforms our simple encoder. This aligns with the observation of Chen et al. (2024) that encoders specifically designed to work on arbitrary homophily levels do not perform well on current GCL methods. We also note a significant improvement in the performance of the homophilic datasets with the decoupled approach, emphasizing its importance in the FOSSIL framework.

Table 4: Ablation study on the encoder method.

| Encoder | Cora | CiteSeer | PubMed | CoAuthor CS | Actor | Chameleon | Squirrel |
|---|---|---|---|---|---|---|---|
| GCN | $\underline{74.72}_{\pm 0.49}$ | $\underline{61.35}_{\pm 0.45}$ | $74.11_{\pm 0.60}$ | $\underline{90.12}_{\pm 0.06}$ | $13.54_{\pm 0.94}$ | $\underline{48.68}_{\pm 0.52}$ | $25.34_{\pm 0.28}$ |
| GCNII | $50.18_{\pm 0.44}$ | $46.45_{\pm 0.40}$ | $68.59_{\pm 0.51}$ | $88.17_{\pm 0.09}$ | $\mathbf{36.10}_{\pm 0.16}$ | $44.75_{\pm 0.33}$ | $\underline{33.71}_{\pm 0.22}$ |
| Decoupled encoder | $\mathbf{80.02}_{\pm 0.26}$ | $\mathbf{68.24}_{\pm 0.34}$ | $\mathbf{79.76}_{\pm 0.48}$ | $\mathbf{94.49}_{\pm 0.04}$ | $\underline{35.61}_{\pm 0.15}$ | $\mathbf{53.03}_{\pm 0.33}$ | $\mathbf{37.13}_{\pm 0.24}$ |

Table 5: Graph perturbation ablation study results on node classification.

| Generator | Cora | CiteSeer | PubMed | CoAuthor CS | Actor | Chameleon | Squirrel |
|---|---|---|---|---|---|---|---|
| Random | $78.28_{\pm 0.35}$ | $65.85_{\pm 0.39}$ | $76.54_{\pm 0.42}$ | $89.40_{\pm 0.27}$ | $28.95_{\pm 1.58}$ | $\mathbf{57.29}_{\pm 1.34}$ | $38.22_{\pm 0.96}$ |
| GAT-F | $74.47_{\pm 0.31}$ | $66.84_{\pm 0.35}$ | $70.25_{\pm 0.51}$ | $92.72_{\pm 0.08}$ | $34.68_{\pm 0.17}$ | $53.69_{\pm 0.30}$ | $34.53_{\pm 0.54}$ |
| F-GAT-E | $75.53_{\pm 0.45}$ | $62.72_{\pm 0.48}$ | $78.33_{\pm 0.42}$ | $90.60_{\pm 0.32}$ | $33.68_{\pm 0.18}$ | $52.68_{\pm 0.82}$ | $33.82_{\pm 0.48}$ |
| GCN-E | $76.95_{\pm 0.32}$ | $67.70_{\pm 0.34}$ | $79.44_{\pm 0.50}$ | $93.49_{\pm 1.93}$ | $34.33_{\pm 0.62}$ | $46.83_{\pm 0.77}$ | $\mathbf{39.43}_{\pm 0.23}$ |
| GAT-E | $\mathbf{80.02}_{\pm 0.26}$ | $\mathbf{68.24}_{\pm 0.34}$ | $\mathbf{79.76}_{\pm 0.48}$ | $\mathbf{94.49}_{\pm 0.04}$ | $\mathbf{35.61}_{\pm 0.15}$ | $53.03_{\pm 0.33}$ | $37.13_{\pm 0.24}$ |

Table 6: Ablation study on fusion the design of the fusion

| Design | Cora | CiteSeer | PubMed | CoAuthor CS | Actor | Chameleon | Squirrel |
|---|---|---|---|---|---|---|---|
| w/o centrality | $\mathbf{80.24}_{\pm 0.25}$ | $67.96_{\pm 0.29}$ | $\mathbf{79.95}_{\pm 0.37}$ | $\mathbf{95.04}_{\pm 0.04}$ | $35.76_{\pm 0.60}$ | $54.24_{\pm 0.48}$ | $36.43_{\pm 0.27}$ |
| w/ degree | $80.02_{\pm 0.26}$ | $68.24_{\pm 0.34}$ | $79.76_{\pm 0.48}$ | $94.49_{\pm 0.04}$ | $35.61_{\pm 0.15}$ | $53.03_{\pm 0.33}$ | $37.13_{\pm 0.24}$ |
| w/ degree normalized-1 | $79.44_{\pm 0.29}$ | $68.77_{\pm 0.27}$ | $79.36_{\pm 0.40}$ | $94.78_{\pm 0.03}$ | $30.48_{\pm 2.00}$ | $53.00_{\pm 0.94}$ | $36.79_{\pm 0.21}$ |
| w/ degree normalized-2 | $79.68_{\pm 0.27}$ | $69.77_{\pm 0.29}$ | $79.70_{\pm 0.35}$ | $94.39_{\pm 0.05}$ | $35.16_{\pm 0.18}$ | $53.94_{\pm 0.31}$ | $36.97_{\pm 0.20}$ |
| w/ pagerank | $79.14_{\pm 0.30}$ | $69.63_{\pm 0.30}$ | $79.45_{\pm 0.42}$ | $94.34_{\pm 0.08}$ | $\mathbf{35.94}_{\pm 0.15}$ | $\mathbf{54.84}_{\pm 0.40}$ | $36.77_{\pm 0.22}$ |
| w/ eigenscore | $79.45_{\pm 0.28}$ | $\mathbf{69.89}_{\pm 0.27}$ | $79.83_{\pm 0.37}$ | $93.90_{\pm 2.66}$ | $36.08_{\pm 0.15}$ | $53.67_{\pm 0.79}$ | $\mathbf{37.38}_{\pm 0.61}$ |

### 5.3.3 Impact of the view generation process

We validate the design of the generator of FOSSIL by testing other alternatives. We first replace the generator with a traditional non-parametric random perturbation scheme (denoted as Random). Specifically, we perform random edge dropping ($\sim 15\%$) to assess the benefit of a learnable view generation. Moreover, we investigate whether to apply the GAT on embeddings (GAT-E), initial features (GAT-F), or an intermediate approach in which we compute attention weights with the features and directly apply them on the embeddings (F-GAT-E). Finally, we replace the GAT with a GCN that is applied to the embeddings (GCN-E). Table 5 shows the results of this ablation study. First, we notice that both the GCN and GAT models outperform the random perturbation. Even though random perturbation presents competitive results, it requires careful fine-tuning of the hyperparameters according to the graph (You et al., 2020). Secondly, we notice that applying the GAT on initial features does not work well compared to applying it on the node embeddings. We believe that applying the GAT on node features in our framework is equivalent to learning a different encoder, which is not aligned with the intended view generation. The intermediate approach (F-GAT-E) also yields competitive results but still underperforms (GAT-E), suggesting that this approach is less aligned with the intended view generation compared to (GAT-E). However, applying the GAT on node embeddings is more aligned with the initial intuition to generate similar views. Finally, we observe that GAT-E provides better results than GCN-E since the attention mechanism provides flexibility during contrastive learning.

### 5.3.4 Ablation on the design of the fusion module

We incorporate the degree information of nodes in our fusion module. Intuitively, this brings useful structural information on nodes inside the graph. To validate this intuition, we explore different alternatives. More precisely, we remove the degree information and test other node centralities: scaling degrees with $N-1$ (normalized-1) or by the sum of degrees (normalized-2), PageRank (Page et al., 1999) and Eigenvector (Burr, 1982) centralities. Table 6 shows the results of this study. We observe that removing the node degree outperforms the degree centrality in some datasets. Nevertheless, we further observe that other global centrality measures like Eigenscore or PageRank outperform the baseline in several cases. Therefore, the degree of centrality offers a good trade-off between complexity and performance.

# 6 Conclusion

In this paper, we introduced a novel subgraph-based contrastive learning framework utilizing the Fused Gromov-Wasserstein distance to measure subgraph similarities. Drawing inspiration from recent advancements in addressing heterophily with GNNs, we implemented a decoupled embedding extraction strategy, rendering our method agnostic to the graph's homophily level. This approach allowed us to independently generate feature and structure perturbations adaptively. Our extensive experimental evaluation demonstrated that our method outperforms or has competitive performance against state-of-the-art frameworks across multiple benchmark datasets with varying levels of homophily. Our findings also highlight the challenges posed by heterophilic datasets. We suggest future research avenues, such as integrating a high-pass feature extractor to enhance performance in such scenarios. Similarly, extending FOSSIL to other downstream tasks such as link prediction or graph classification deserves further research.

### Acknowledgments

We thank the anonymous reviewers for their suggestions and feedback. This work was supported in part by ANR (French National Research Agency) under the JCJC projects GraphIA (ANR-20-CE23-0009-01), and DeSNAP (ANR-24-CE23-1895-01).

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

# A    Algorithm for FGWD

---

**Algorithm 1** Bregman Alternated Projected Gradient (BAPG) for FGWD

---
$\mathbf{P}^{(0)} \leftarrow \boldsymbol{\mu}\boldsymbol{\nu}^{\top}$
$\Delta\mathbf{P} \leftarrow \infty$
$i \leftarrow 1$
**while** $(i < T)$ **And** $(\Delta\mathbf{P} > \epsilon)$ **do**
   // Row updating
   $\mathbf{G} \leftarrow \alpha\mathbf{M} + 2(1-\alpha)\underline{\mathbf{L}}(\mathbf{C}_1, \mathbf{C}_2) \otimes \mathbf{P}^{(i-1)}$ // Gradient of the objective function of Eq. (1) in $\mathbf{P}^{(i-1)}$
   $\tilde{\mathbf{P}} \leftarrow \mathbf{P}^{(i-1)} \odot \exp(-\mathbf{G}/\beta)$
   $\tilde{\mathbf{P}} \leftarrow \mathrm{diag}(\frac{\boldsymbol{\mu}}{\tilde{\mathbf{P}}\mathbb{1}_m})\tilde{\mathbf{P}}$

   // Column updating
   $\mathbf{G} \leftarrow \alpha\mathbf{M} + 2(1-\alpha)\underline{\mathbf{L}}(\mathbf{C}_1, \mathbf{C}_2) \otimes \tilde{\mathbf{P}}$ // Gradient of the objective function of Eq. (1) in $\tilde{\mathbf{P}}$
   $\mathbf{P}^{(i)} \leftarrow \tilde{\mathbf{P}} \odot \exp(-\mathbf{G}/\beta)$
   $\mathbf{P}^{(i)} \leftarrow \mathbf{P}^{(i)} \mathrm{diag}(\frac{\boldsymbol{\mu}}{\mathbf{P}^{(i)\top}\mathbb{1}_n})$
   $\Delta\mathbf{P} \leftarrow \|\mathbf{P}^{(i)} - \mathbf{P}^{(i-1)}\|$
   $i \leftarrow i + 1$
**end while**

---

# B    Evaluation on Larger Graphs

We compare FOSSILv2 with some baselines on two large-scale graphs: OGBN-Arxiv and OGBN-Proteins, containing 1M and 30M edges, respectively (Hu et al., 2020a). For OGBN-Proteins, we extract a subgraph of 1M edges due to computational resource limitations. To this end, we use the `NeigbhorLoader` loader of PyG to sample, for some nodes, 18 neighbors in each layer of its 3-hop neighborhood (Hamilton et al., 2017). Any number above 18 results in a number of edges that lead to a memory overflow with our method and all baselines. We show the results in Table 7. Some baselines present in Table 2 result in memory overflow, therefore, they are not included in this experiment. We do not show results for DSSL in OGBN-Proteins because it does not work for graphs where we have disconnected regions. We observe that our method is competitive against the baselines.

Table 7: Test accuracy on the node classification on large-scale graphs. The best results are highlighted in **bold**, while the second best-performing methods are underlined.

| Method | OGBN-Arxiv | OGBN-Proteins |
|---|---|---|
| FS-GCN | $30.75_{\pm 0.32}$ | $88.67_{\pm 0.01}$ |
| FS-MLP | $23.16_{\pm 0.58}$ | $88.70_{\pm 0.01}$ |
| GSC | $35.19_{\pm 0.36}$ | $88.87_{\pm 0.01}$ |
| GSC+Cheb | $34.97_{\pm 0.53}$ | $88.64_{\pm 0.01}$ |
| DSSL | $\underline{58.36}_{\pm 2.69}$ | $-$ |
| BGRL | $54.16_{\pm 0.30}$ | $\mathbf{89.32}_{\pm 0.01}$ |
| Subg-Con | $\mathbf{60.82}_{\pm 0.08}$ | $\underline{89.04}_{\pm 0.01}$ |
| FOSSILv2 | $56.22_{\pm 0.12}$ | $88.63_{\pm 0.01}$ |

# C    Sensibility Analysis

One of the most important hyperparameters of FOSSIL is $\alpha$, which dictates how much our model should focus on each source of information of the input graph, namely, node features and graph structure. To assess how sensitive our framework is to the value of $\alpha$, we vary its value and see how the performance changes.

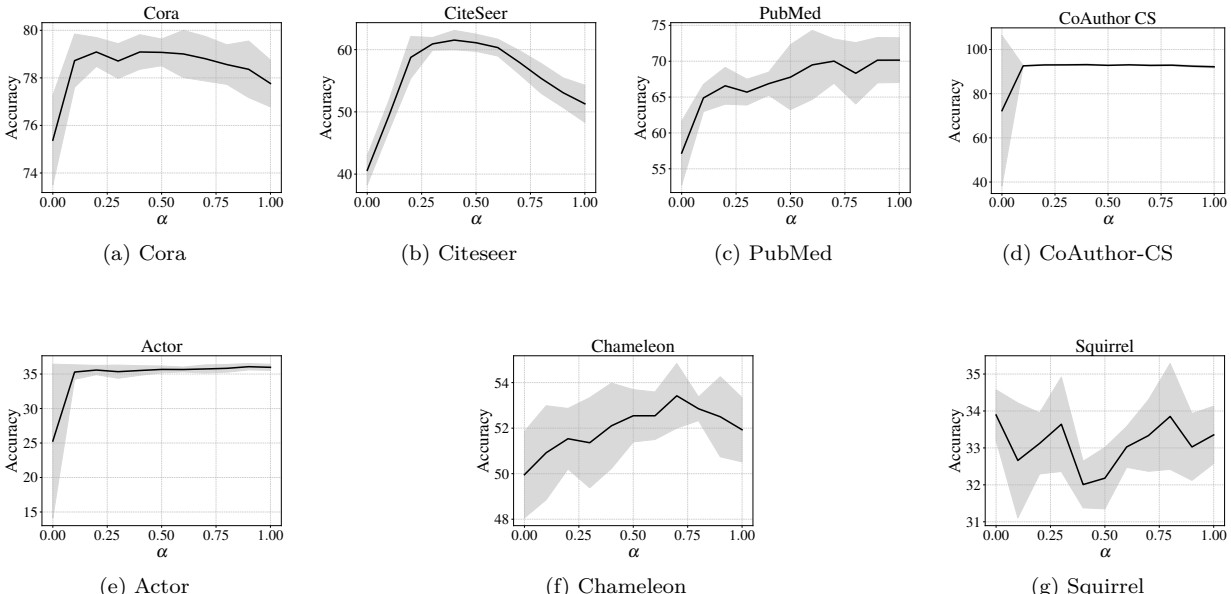

Figure 3: Variation of the accuracy for several datasets w.r.t. $\alpha$.

Specifically for each real-world dataset in our experiments, we fix the other hyperparameters and vary $\alpha$ within the set $\{[0 : 0.1 : 1]\}$. For each value of $\alpha$, we evaluate our method on 5 seeds and report the mean and standard deviation. Figure 3 shows the sensibility analysis results. We notice that the performance of our method can significantly change depending on the value of $\alpha$. Overall, for homophilic graphs, the best value of $\alpha$ tends to be close to 0.5. This suggests that node features and graph structure are equally important for the encoder to characterize the graph. However, for heterophilic graphs, the best value of $\alpha$ tends to be close to 1. A plausible explanation is that even though the graph structure is important, the GCN encoder is not able to correctly exploit the structure of heterophilic graphs due to its intrinsic homophily assumption; our framework will rely more on node features. Hence, our encoder will converge to an MLP with a value of $\alpha$ close to 1. This further supports our idea to include a feature extractor that is more suited to heterophilic graphs for future research.

