# OpenReview forum: "A Fused Gromov-Wasserstein Approach to Subgraph Contrastive Learning"
_TMLR — Accepted by TMLR_

### Review · Reviewer_STEg · 2024-09-20

**Summary Of Contributions:**

This paper introduces a new contrastive learning framework for graph representation learning, termed Fused Gromov-Wasserstein Subgraph Contrastive Learning (FOSSIL). The advantage of FOSSIL is three-fold:

1. FOSSIL adopts Fused Gromov-Wasserstein Distance (FGWD) in the training loss, which can integrate both structural and feature similarities in a unified contrastive learning framework.

2. FOSSIL involves a decoupled embedding design to work well across both homophily and heterophily graphs.

3. FOSSIL dynamically generates subgraph augmentations by Graph Attention Network (GAT). This provides a learnable and adaptive approach to data perturbation.

Experiment results are provided to validate the benefit of FOSSIL.

**Audience:**

Yes

**Claims And Evidence:**

Yes

**Requested Changes:**

Add a formal problem definition of the problem definition of contrastive graph learning, and briefly introduce its motivation (e.g., how it can be used in downstream problems).

In Eq. (5), the author introduces an MLP of input dimension $\mathbb{R}^{N\times D\times N\times D\times N}$, which is too high. Please check.

**Strengths And Weaknesses:**

### Strength

FOSSIL captures both node features and graph structure simultaneously. This is a significant strength as previous methods often treat these aspects separately, leading to suboptimal performance.

Despite good experimental results, the paper further addresses potential scalability issues by proposing FOSSILv2. FOSSILv2 reduces the memory complexity of the node-level contrastive loss so that the method can handle larger graphs.

### Weakness

The paper is generally well-written. However, the formal problem statement is missing. The paper only outlines general issues in contrastive graph learning, and then directly introduces the methodology. It does not provide an explicit and concise problem definition. Readers without much background knowledge of contrastive graph learning will find it difficult to follow.

---

> ### Author Response · Authors · 2024-11-20
> **Introducing a formal definition of graph contrastive learning and fixing the fusion layer MLP's description**
>
> We thank reviewer **STEg** for the important remarks. Our response is organized in a point-by-point format, as outlined below:
> - To address the lack of a formal definition of graph contrastive learning, we included a section in which we define the theoretical goal of graph contrastive learning, how it is performed in practice, and how the learned representations are transferred to graph-related downstream tasks;
> - We also rectified the error spotted in the description of the fusion module MLP in now Eq. (8) of the revised manuscript, the correct definition is $\psi: \mathbb{R}^ D\times\mathbb{R}^D\times\mathbb{R} \to \mathbb{R}$.

---

> > ### Comment · Reviewer_STEg · 2025-01-30
> >
> > Thank the authors for the response. My comments are mostly minor, The added section on GCL is clear. I have no further comments.

---

### Review · Reviewer_TVUs · 2024-12-07

**Summary Of Contributions:**

The authors identify that to improve SSL on graphs, the CL loss needs to incorporate both feature information usually captured by Wasserstein Distance (WD) and subgraph structural information usually captured by Gromov-Wasserstein Distance (GWD). The authors also point out that heterophily posts a significant challenge when generating positive-negative pairs. To fill this gap, the authors propose the FOSSIL - Fused Gromov-Wassertein Subgraph Contrastive Learning:
1. An learnable linear combination of WD and GWD to balance the structural information and feature information as a contrastive loss function.
2. A 2-stage training strategy for GNNs to tackle heterophily: (1) Use Identity matrix as adjacency to transform just the node embeddings (MLP behavior) and (2) Without adding self-loops (usually done in MPNNs), do message-passing. Finally fuse (1) and (2) using linear combination determined by an MLP.

They conduct extensive experiments on self-supervised node-classification task and evaluate their design choice, which show that FOSSIL shows competitive performance, especially on heterophilic graphs.

**Audience:**

Yes

**Broader Impact Concerns:**

No concerns

**Claims And Evidence:**

Yes

**Requested Changes:**

Please address the concerns mentioned in the weaknesses and adjust the submission. Additionally, W7 is very concerning. It is unclear why GWD fails consistently on all kinds of graphs. The ablation does not help to understand the imporance of FGWD. I suggest to perform the following ablation: Using standard GCN, ablate on subgraph similarity metric. This is help us isolate the benefit of FGWD.

**Strengths And Weaknesses:**

Strengths:
1. The authors motivate the need for SSL on graphs and existing gaps using solid literature review.
2. Notations and Problem Setup is very clear.
3. Proposing the Fused GWD as a loss function is novel. This is a simple and effective solution to fill the gap in existing Subgraph CL methods.

Weaknesses:
1. The authors claim that CL on graphs is hard because generating positive pairs is hard without any references.
2. The authors claim that their architecture is robust to various homophily levels within the graph dataset. It is not clear whether the authors refer to distinct datasets with different homophily levels or graphs having different homophily levels within the same dataset. If it is the latter, there needs to be more experiments. If not, it needs revision.
3. The authors mention the transformation through first encoder as "linear projection" which is incorrect. It has a non-linearity as shown in Equation (6), which does not make it a linear projection.
4. It is not clear how "Fusion" is different from a standard GCN/MPNN framework (with self-loops). The process is similar to scaling the aggregated messages.
5. Subsection 5.3.1 and 5.3.2 have the same headings "Impact of decoupled node embeddings". This is a typo.
6. The authors do not explain what the "Overall" column is in any of their results.
7. In Table 3, the performance of GWD is alarming. This means $\alpha=0$. The authors do not explain why GWD performs bad on homophilic graphs, where the subgraph structure information should actually be helpful.

---

> ### Author Response · Authors · 2025-01-14
>
> We would like to clarify that the Fused Gromov-Wasserstein Distance is not a “learnable combination of the Wasserstein Distance and Gromov-Wasserstein Distance”. In fact, it is an interpolation of the two distances, but the interpolation coefficient $\alpha$ is a hyperparameter of our framework.
>
> **W1.**
> In the manuscript, we provided several references pointing to the fact that it is hard to define a proper augmentation strategy in the context of graph contrastive learning. Please find the references bellow.
>
> **W2.**
> We are referring to different graphs with different levels of homophily. We understand that it may be confusing but when using the term "dataset", we are referring to graphs. Please notice that Cora, CiteSeer, PubMed CoAuthor-CS, OGBN-Arxiv, Actor, Chameleon, and Squirrel are referred to in the literature as graph datasets, although they are graphs.
>
> **W3.**
> We corrected this error in the new manuscript.
>
> **W4.**
> Our encoder is not a GCN. Our encoder is an MPNN where we aggregate information from the neighborhood of each node and we have an additional MLP. How these two encoders are merged is learned separately.
>
> **W5 and W6.** We addressed the issues in the new manuscript.
>
> **W7.** We noticed that the regularizer $\beta$ to compute the FGWD was too low for the BAPG algorithm to converge. We addressed the issue by increasing the $\beta$ and ran the experiment again.
>
>
> ### References
>
> [1] Li et al. Augmentation-Free Graph Contrastive Learning of Invariant-Discriminative Representations. IEEE Transactions on Neural Networks and Learning Systems. 2022
>
> [2] Han et al. Generative Subgraph Contrast for Self-Supervised Graph Representation Learning. ECCV 2022

---

> > ### Comment · Reviewer_TVUs · 2025-01-18
> >
> > Dear Authors,
> >
> > Thank you for addressing all the concerns. I am satisfied with your responses. There is a new formatting issue in Section 2.2 - a huge gap between the subheading and the paragraph. Please change it.

---

### Review · Reviewer_NhfX · 2024-12-07

**Summary Of Contributions:**

The paper presents a method that integrates node and subgraph level contrastive learning to be used a pretext task that generates representations later used for node level tasks. The architecture consists of an encoder, a generator which is responsible for generating perturbed views, and a fusion and subgraph sampling method that generates representations used in two contrastive learning losses, respectively for node and subgraph level.

**Audience:**

Yes

**Claims And Evidence:**

No

**Requested Changes:**

### Critical:
1. Can you please clarify the need of each component? Mainly:

    a. Related to what was discussed in the weaknesses. Why do you need to do additional message passing steps for obtaining perturbations? Can we only do for instance GAT at the feature level (or maybe just add some noise since we already have an embedding)? Note that these should be baselines for section 5.3.3.

   b. Why do we need to remove the self loop in the gcn used to get the structure embedding?

2. Results for ogbn-arxiv are missing from Tables 2,3,4,5 (which contain the baselines)
3. The datasets used are a bit outdated and also small. I think the authors should experiment more with the ogb datasets, and minimally  include ogbn-products and ogbn-proteins, which are medium scale datasets.

### Nice to have:
1. What is the impact of including the node degree in the fusion model? This part also feels a bit unmotivated, so at least having an ablation study on it would strengthen the paper.
2. On a more general note, why only testing on node prediction tasks? Does it mean that we need a pretext task for each graph task, instead of a single pretext task that can be used for many graph tasks (node prediction, graph prediction, link prediction)?
3. There are many works to sample subgraphs in subgraph gnns ([1,2,3] for instance), so please add these references. Also, I think it would be interesting to consider these more convoluted and generally better performing sampling strategies.

### Minor issues:
1. citet/citep is not used correctly in page 3 (Section 2.2)
2. equation pointer in page 8 is broken (i.e., it says equation ??)
3. title of section 5.3.1 is wrong, as that is the title of section 5.3.2

[1] Qian et al. Ordered Subgraph Aggregation Networks

[2] Kong et al. MAG-GNN: Reinforcement Learning Boosted Graph Neural Network

[3] Bevilacqua et al. Efficient Subgraph GNNs by Learning Effective Selection Policies

**Strengths And Weaknesses:**

### Strengths
1. The research topic is important, as designing effective pretext tasks can potentially reduce the number of required labeled data.
2. The math is easy to understand, and the notation is clear.

### Weaknesses
1. The paper is very hard to follow, mainly because the method is not introduced until page 5. To improve readability, the authors should provide a high-level intuitive description of the method earlier in the paper. Delaying the overview and main method description until later pages makes it challenging to follow.
2. The components of the architecture feel unmotivated. I think the authors should investigate more the need for certain components. For instance, the use of an additional GNN (GAT or GCN in the ablation study) to generate perturbations could be better motivated. The generator GNN, which takes the encoder output and performs additional message passing, effectively increases the receptive field for each node, resulting in representations that are not simple perturbations of the encoder outputs (as intended) but rather have a deeper view. This raises questions about whether the generated representations align with the intended objective.
3. Sampling the subgraph breaks the equivariance of the architecture. I think the authors should at least discuss this point. I also believe that this would be more problematic if the datasets were not transductive.

---

> ### Author Response · Authors · 2025-01-14
>
> **W1.**
> We understand your concerns about the redaction of the manuscript. We have done our best to make our message clear and structured. This requires properly defining each notion and all the notations that are used in the paper. When we include the abstract, introduction, and related works section to all of that, we don’t see any other way to introduce the method earlier than page 5.
>
> **W2.**
> We added several additional ablation studies on the loss, the encoder design, generator design to validate our choices. Please find them in Tables 3, 4, 5, and 6 in the revised manuscript.
>
> **W3.**
> Our method does not break equivariance. First, our encoder is, in fact, permutation equivariant as a sum of two permutation equivariant operations (please see equation (5) in the revised manuscript). Secondly, we sample subgraphs after encoding the nodes. Please note that our subgraph sampling is nothing similar to [2,3,4]. These works sample subgraphs before repetitively applying the same encoder to each subgraph. This will indeed break equivariance. In contrast, our method applies the encoder on the full graph only one time, and then sample subsets of nodes in order to make subgraphs.
>
> **R1.**
> We added additional ablation studies.
> - On the way we obtain perturbations, as suggested, we tried applying the GAT model on the features rather than the embeddings. We observed a decrease in performance compared to applying it to the embeddings. We believe that applying the GAT on the embeddings is more aligned with our intuition. Let $f$ be the encoder and $g$ the GAT. Essentially, for a sample $x$, we are creating its augmented view by applying g to its embeddings. Concretely, $(f(x), g(f(x)))$ is considered as our positive pair and $g$ is trained so that $g(f(x))$ is similar to $f(x)$ and dissimilar to $f(x’)$ ($x’ \neq x)$. When we instead apply $g$ to $x$ and use $(f(x), g(x))$ as our positive pair, then it is like we are using another encoder and $f(x)$ is not involved in creating a similar view to itself. $g$ also doesn't get any gradient from $f$ to adapt its view generation.
> - (i) By feature embedding, we mean the embeddings of nodes computed independently from each other. This ignores the structure of the graph and is equivalent to applying an MLP. (ii) By structure embedding, we mean the embeddings aggregated in the neighborhood of each node (excluding itself). Therefore, we need to exclude self-loops to avoid including nodes in their own neighborhood.  This approach is inspired by the H2GCN [1] model and should improve the results on heterophilic graphs according to H2GCN [1].
>
> **R2.**
> We added OGBN-Arxiv in Table 2 for comparison to other baselines. However, as FOSSIL cannot run on it with our computational resources, we have not included it in the other tables. This dataset has been mainly added to assess the scalability (in terms of memory usage) of our method. Moreover, it is interesting to notice that most of our state-of-the-art baselines also do not run on it.
>
> **R3.**
> We mainly used the most common datasets used in prior works on self-supervised learning on graphs. Moreover, in this work, we are interested in node-attributed graphs and OGBN-Proteins is not a node-attributed graph.
>
> **R4.**
> We added an ablation study on the choice of adding the node degree information in the fusion module. More generally, the idea is to give some information about the connectivity of nodes inside the graphs to assist the fusion module in computing the $\lambda$ values. This information can essentially be any graph centrality metric. We tested different centralities such as degree, PageRank, and eigenvector centrality. Our results suggest that adding a centrality score improves the performance of the fusion layer.
>
> ### References
> [1] Zhu et al. Beyond homophily in graph neural networks: Current limitations and effective designs. NeurIPS 2020
>
> [2] Qian et al. Ordered Subgraph Aggregation Networks
>
> [3] Kong et al. MAG-GNN: Reinforcement Learning Boosted Graph Neural Network
>
> [4] Bevilacqua et al. Efficient Subgraph GNNs by Learning Effective Selection Policies

---

> > ### Comment · Reviewer_NhfX · 2025-01-21
> > **Reply to the authors**
> >
> > I would like to thank the authors for their reply. In the following, I will reply to the main points.
> >
> > 1. I still think having the method only appearing in page 5 significantly impacts readability. The introduction only generally hints the method, and the related work and the background section feel disconnected, because it is not highlighted what is different from the proposed method. I understand that stressing differences and similarities with existing works require effort, but this is fundamental for the flow of the paper.
> >
> > 2. I still do not understand why one should use an additional GNN to generate perturbations, because, as I was saying, additional message passing allows nodes to see more distant neighbors. As a result, the resulting representations are not simple perturbations (which is the objective according to the authors). Following your math, why can't we do $(f(x), g(f(x))$, but with $g$ being a GAT at the feature level **applied on $f(x)$**? In other words, perturbations still see $f(x)$ as input, but $g$ is not a GNN (but rather an attention layer).
> >
> > 3. Any sampling strategy by design breaks the equivariance *after the sampling layer*. This is because even passing the same graph will potentially lead to different sampled subgraphs which, in turn, will give a different output of the contrastive loss while learning. Note that this is not a weakness, I was suggesting to discuss this point in the paper, and maybe refer to the suggested papers in the related work, as their proposed subgraph sampling strategies could be used to sample subgraphs.
> >
> > 4. Note that from the ogb github repo, node features are obtained for ogbn-proteins from the edge features. Also ogbn-products has node features. For future versions, I would encourage the authors to add these medium sized datasets.

---

> > > ### Author Response · Authors · 2025-02-04
> > >
> > > 1. We added paragraphs in the related works to show how our method position compared to the cited works. We hope this gives a sooner introduction of our method.
> > >
> > > 2. We performed a new experiment based on the view generation approach the reviewer suggested, namely computing GAT attention weights with the original features and then applying them to the embeddings. We refer to this model as F-GAT-E. The results are given in Table 5. As we observe, we have a lower performance compared to our method (GAT-E). We think that what the reviewer proposed is still a message-passing scheme. An MPNN can be expressed as follows: $\mathbf{H} = \sigma(\mathbf{SXW})$ with $\mathbf{S}$ a shift operator, $\mathbf{W}$ a linear projection layer (the function that computes the message), and $\mathbf{X}$ the input node features. From our understanding of your suggestion, we compute the shift operator $\mathbf{S}$ as done in GAT [1] with $\mathbf{X}$ and then apply it to the node embeddings. Formally, let $\mathcal{G} = (\mathbf{X}, \mathbf{A})$ a graph and f the encoder. We compute $\mathbf{\hat{H}} = \sigma(\mathbf{SH})$ where $\mathbf{H} = f(\mathbf{X}, \mathbf{A})$ and $\mathbf{S} = \text{GAT-Attention}(\mathbf{X}, \mathbf{A})$. Notice that we get a message passing scheme in which $\mathbf{W}$ is the identity matrix. If the reviewer thinks that this is aligned with the intended view generation, our approach should be as well because they are both message-passing schemes.
> > >
> > > 3. Thanks for the suggested references. In fact, their data-driven subgraph sampling can improve our method in future works. We discuss that in section 4.5. However, these works [2, 3,4] don’t explicitly discuss the fact that subgraph sampling breaks equivariance. Speaking of equivariance, we mostly think of it at the encoder level. That is, is our encoder permutation-equivariant? As a linear combination of two permutation-equivariant architectures (a GNN and an MLP), our encoder is, in fact, permutation-equivariant. While we are still not sure about the problem with equivariance, we understand your point that sampling subgraphs can lead to different outputs of the contrastive loss. We also shortly discuss this point in the new manuscript.
> > >
> > > 4. We understand the importance of comparing the baselines and FOSSIL in several datasets. However, we estimate that the time required to run our method and the other baselines for ogb-proteins dataset is one week with our current resources, which is not possible given the timeframe of TMLR. We believe that running FOSSIL in OGBN-Arxiv is sufficient to show the scalability properties of our method, which is the most important aspect in our case
> > >
> > > ### References
> > >
> > > [1] Velickovic et al. Graph Attention Networks. ICLR 2018
> > >
> > > [2] Qian et al. Ordered Subgraph Aggregation Networks
> > >
> > > [3] Kong et al. MAG-GNN: Reinforcement Learning Boosted Graph Neural Network
> > >
> > > [4] Bevilacqua et al. Efficient Subgraph GNNs by Learning Effective Selection Policies

---

> > > > ### Comment · Reviewer_NhfX · 2025-02-05
> > > >
> > > > Thank you for the reply.
> > > >
> > > > I am not sure you have been answering the intended question, but I may be wrong. My initial point was that the generator GNN (which can be GAT) should not use $\mathbf{A}$ (and can therefore only use $\mathbf{I}$). This is because the generator GNN should not do additional message passing. Can you please clarify if you tried this variant and how is this called in section 5.3.3?
> > > >
> > > > Yes your encoder is certainly equivariant. It is the sampling that breaks equivariance.
> > > >
> > > > Can you please add the requested experiment on ogbn-proteins? I think waiting one additional week is fine, given that this experiment was requested on December 7th. OGBN-Arxiv is not sufficient as arxiv is considered a small dataset in the ogb website (1,166,243 edges) while proteins is considered medium (39,561,252 edges, therefore around 33 times larger).

---

### Review · Reviewer_hh1Z · 2024-12-09

**Summary Of Contributions:**

The paper introduces Fused Gromov-Wasserstein Subgraph Contrastive Learning (FOSSIL), a self-supervised framework that captures both feature and structural similarities in graphs. Designed to work across varying levels of homophily, FOSSIL employs a decoupled encoding strategy and integrates node- and subgraph-level contrastive losses. Experimental results demonstrate good performance on diverse graph datasets compared to state-of-the-art methods.

**Audience:**

Yes

**Claims And Evidence:**

No

**Requested Changes:**

See weaknesses and minor issues.

**Strengths And Weaknesses:**

### Strengths:
- The problem addressed is interesting, especially the applicability to heterophilic datasets.
- Notation and experimental design are clear.

### Weaknesses:
- The paper is not easy to follow:
   1) The method appears only on page 5, while an improved version (namely FOSSILv2) appears on page 12 in the Limitations section.
   2) There is no clear and strong motivation behind each architectural choice. Authors should better clarify the rationale behind their choices. Specifically:

         a) Why is each component necessary?

         b) Why use GCN and GAT as backbones instead of GNNs designed for heterophilic graphs?

         c) Why do the encoder and the generator use shared weights?

         d) Why is the proposed decoupled encoder is more suitable for heterophilic graphs than current methods (including those designed to work in the heterophilic setting)?
   3) The authors split the Encoder section (Sec 4.2) into 2 subsections, which makes the text feel redundant and overcomplicated.
   4) The figure and the text could be better linked.


- The authors claim that current methods have GNN encoders that struggle with heterophilic datasets. Since the authors propose an encoder that is based on GCN, it is surprising that no comparison with methods specifically designed for heterophilic graphs, such as Geom-GCN [1], GCNII [2], or H_2GCN [3], is found to justify their choice.
- OGBN-Arxiv is not cited in the list of cited datasets in Section 5, and it is missing from Tables 2, 3, 4, and 5.
- Section 5.1 lacks important hyperparameters, such as the number of layers, the hidden dimension, optimizer, and number of epochs,  which limit the reproducibility of the experiments.
- The authors split the development set into training and validation by considering 20 nodes per class for the training set and the rest for the validation set. Is this setting the same as in [4]? If so, the paper should be cited.
- The authors should consider more recent homophilic and heterophilic tasks, such as [5], to strengthen their claim that FOSSIL works well with both homophilic and heterophilic graphs.


### Minor issues:
- Although it is clear, the acronym GNN is never defined.
- Page 3: citep is not used correctly for GPR-GNN, ChebNetII, and PolyGCL.
- Page 5: “the second [neural network] implements a GNN (Kipf & Welling, 2017)” . Is GNN a typo? Should GNN be replaced with GCN?
- The titles of sections 5.3.1 and 5.3.2 are the same.
- Table 1: a comma is missing in the Squirrel’s number of nodes.

--------
[1] Pei et al. Geom-GCN: Geometric Graph Convolutional Networks. ICLR 2020

[2] Chen et al. Simple and Deep Graph Convolutional Networks. ICML 2020

[3] Zhu et al. Beyond homophily in graph neural networks: Current limitations and effective designs. NeurIPS 2020

[4] Shchur et al. Pitfalls of Graph Neural Network Evaluation. Relational Representation Learning Workshop, NeurIPS 2018

[5] Luan et al. The Heterophilic Graph Learning Handbook: Benchmarks, Models, Theoretical Analysis, Applications and Challenges. 2024

---

> ### Author Response · Authors · 2025-01-14
>
> **W1.**
> - On the redaction of the paper, we have taken into account the reviewer’s concerns and introduced FOSSILv2 earlier in the revisited version of the manuscript (page 8). However, we are afraid we can’t introduce our method earlier than page 5 as we need to properly define every notion used in the manuscript. Moreover, we have rewritten section 4.2 (now section 4.3) to make it more concise.
> We added additional ablation studies to assess the necessity of each component when applicable.
> - For question b), we tested other GNNs designed to work on arbitrary homophily levels. We refer the reviewer to Table 4 where we report the results obtained when replacing the GCN encoder with a GCNII, as suggested. Interestingly, it underperforms our baseline. This observation aligns with those of PolyGCL which explains that these more homophily-agnostic encoders usually underperform encoders that rely on the homophily assumption.
> - For question c), we would like to emphasize the fact that it is essentially the same encoder and generator. What is changing is the input adjacency matrix for the two different embeddings, namely feature and structure embedding. Therefore, they naturally share the weights of the base encoder and generator.
> - For question d), please note that we are not claiming that our encoder is better than current encoders specifically designed for heterophilic graphs. These encoders in fact work well on heterophilic graphs but in a supervised setting. However, they underperform homophily-sensitive GNNs like GCN in current self-supervised learning frameworks as highlighted by [1]. This is something we also observe in our experiments.
>
> **W2.** We performed an ablation study where we replaced the encoder with a GCNII. We observe that it underperforms our encoder. Moreover, this aligns with the observations of [1].
>
> **W3.** We added OGBN-Arxiv to the list of cited datasets and added it in Table 2. We also added FOSSILv2 in Table 2 and removed Table 6. However, as FOSSIL can’t run on OGBN-Arxiv with our computational resources, it is not included in the other tables.
>
> **W4.** We added the lacking hyperparameters. Namely the number of layers, hidden dimensions, activation functions of the encoder, and the optimizer. Moreover, the code has been released for reproducibility.
>
> **W5.** The references to the splits have been added in the revisited manuscript.
>
> **W6.** We understand the importance of evaluating our method on more recent graph datasets. However, our choice to test on well-established benchmark graphs ensures comparability with state-of-the-art methods and provides a consistent context for evaluation.
>
>
> ### References
>
> [1] "this simple idea results in performance degradation in self-supervised settings", Chen et al., "PolyGCL: GRAPH CONTRASTIVE LEARNING via Learnable Spectral Polynomial Filters", ICLR 2024

---

> > ### Comment · Reviewer_hh1Z · 2025-01-20
> >
> > Dear Authors,
> >
> > Thank you for your reply. Although I still believe that even newer datasets can ensure comparability, I am satisfied with your answers.
> >
> > To give the authors an example, it is possible to provide 43 different baselines for the Roman-empire, Amazon-ratings, Minesweeper, Tolokers, and Questions datasets by considering [1-3].
> >
> > [1] Finkelshtein et al. Cooperative Graph Neural Networks. ICML 2024
> >
> > [2] Platonov et al. A critical look at the evaluation of GNNs under heterophily: Are we really making progress?. ICLR 2023
> >
> > [3] Muller et al. Attending to graph transformers. TMLR 2024

---

### Author Response · Authors · 2025-01-14

We express our gratitude to the reviewers for their thoughtful and constructive feedback. We appreciate their recognition of some strengths of our work and the relevance of our problem. In particular, the reviewers found that the notation and experimental designs are clear (several reviewers), “The problem addressed is interesting, especially the applicability to heterophilic datasets” (reviewer **hh1Z**), “The research topic is important, as designing effective pretext tasks can potentially reduce the amount of required labeled data” (reviewer **NhfX**), “The authors motivate the need for SSL on graphs and existing gaps using a solid literature review” (reviewer **TVUs**), and “Despite good experimental results, the paper further addresses potential scalability issues by proposing FOSSILv2” (reviewer **STEg**).

We have addressed each reviewer’s comment in a detailed manner. We focused on addressing concerns about the design choices, clarifying points of uncertainty, resolving misunderstandings that may have arisen, and as much as possible, simplifying the redaction of the manuscript. Our responses are organized in a point-by-point format. A summary of these responses is listed as follows:
- We added additional ablation studies to strengthen our design choices.
- We did our best to address the redaction issues.
- We address some errors and typos raised by the reviewers.

---

### Decision · Action_Editor_h4yy · 2025-02-14

**Recommendation:** Accept with minor revision

**Comment:**

Please clarify the remaining concerns raised by reviewer NhfX. Specifically, quoting from the discussion with the Reviewer:

1."I am not sure you have been answering the intended question, but I may be wrong. My initial point was that the generator GNN (which can be GAT) should not use  (and can therefore only use ). This is because the generator GNN should not do additional message passing. Can you please clarify if you tried this variant and how is this called in section 5.3.3?" -- Please address this question in your camera ready revision.

2."Yes your encoder is certainly equivariant. It is the sampling that breaks equivariance." -- Please add a discussion that reflects this point in the camera ready version.

3."Can you please add the requested experiment on ogbn-proteins? I think waiting one additional week is fine, given that this experiment was requested on December 7th. OGBN-Arxiv is not sufficient as arxiv is considered a small dataset in the ogb website (1,166,243 edges) while proteins is considered medium (39,561,252 edges, therefore around 33 times larger)." -- Please add results on the requested dataset.

**Audience:**

All reviewers agree that this work fits to TMLR.

**Claims And Evidence:**

Reviewers agree that the claims are mostly solid and are mostly backed up by evidence.